# Multi-model functionalization of disease-associated *PTEN* missense mutations identifies multiple molecular mechanisms underlying protein dysfunction

Kathryn L. Post[1,2,6], Manuel Belmadani [3,4,6], Payel Ganguly[1,2,6], Fabian Meili [1,2,6], Riki Dingwall[1,2,6], Troy A. McDiarmid[1,5,6], Warren M. Meyers [1,2], Caitlin Herrington[2], Barry P. Young [2], Daniel B. Callaghan[3,4], Sanja Rogic [3,4], Matthew Edwards[1,2], Ana Niciforovic[1,2], Alessandro Cau[1,2], Catharine H. Rankin [1,5], Timothy P. O'Connor[1,2], Shernaz X. Bamji[1,2], Christopher J. R. Loewen[2], Douglas W. Allan[1,2], Paul Pavlidis [1,3,4,7✉] & Kurt Haas[1,2,7✉]

Functional variomics provides the foundation for personalized medicine by linking genetic variation to disease expression, outcome and treatment, yet its utility is dependent on appropriate assays to evaluate mutation impact on protein function. To fully assess the effects of 106 missense and nonsense variants of PTEN associated with autism spectrum disorder, somatic cancer and PTEN hamartoma syndrome (PHTS), we take a deep phenotypic profiling approach using 18 assays in 5 model systems spanning diverse cellular environments ranging from molecular function to neuronal morphogenesis and behavior. Variants inducing instability occur across the protein, resulting in partial-to-complete loss-of-function (LoF), which is well correlated across models. However, assays are selectively sensitive to variants located in substrate binding and catalytic domains, which exhibit complete LoF or dominant negativity independent of effects on stability. Our results indicate that full characterization of variant impact requires assays sensitive to instability and a range of protein functions.

[1] Djavad Mowafaghian Centre for Brain Health, University of British Columbia, Vancouver, Canada. [2] Department of Cellular and Physiological Sciences, University of British Columbia, Vancouver, Canada. [3] Department of Psychiatry, University of British Columbia, Vancouver, Canada. [4] Michael Smith Laboratories, University of British Columbia, Vancouver, Canada. [5] Department of Psychology, University of British Columbia, Vancouver, Canada. [6]These authors contributed equally: Kathryn L. Post, Manuel Belmadani, Payel Ganguly, Fabian Meili, Riki Dingwall, Troy A. McDiarmid. [7]These authors jointly supervised this work: Paul Pavlidis, Kurt Haas. ✉email: paul@msl.ubc.ca; kurt.haas@ubc.ca

Autism spectrum disorder (ASD) is the most common genetic neurodevelopmental disorder, characterized by impaired social communication and restricted, repetitive behaviors[1]. Headway in understanding ASD pathophysiology has been stymied by tremendous phenotypic heterogeneity and complex genetics across individuals. Conclusively linking genes to ASD is challenging since sequencing efforts have identified mutations in hundreds of genes from individuals with the disorder[2–7]. Confidence of gene association to ASD is enhanced by identification of de novo, likely gene disrupting (LGD) mutations; however, the most common de novo mutations identified in individuals with ASD are missense (MS) variants of unknown significance (VUS)[2,7]. A complete understanding of ASD genetics and their link to pathophysiology requires deciphering how these mutations impact protein function, and how protein dysfunction affects neural circuits and behavior[8].

The lipid and protein phosphatase, PTEN (phosphatase and tensin homolog deleted on chromosome 10), has well-established links to ASD, somatic cancers, and the cancer-predisposing PTEN hamartoma syndrome (PHTS)[9]. However, while LGD mutations in PTEN have been found in individuals with ASD, the majority of variants identified are MS VUS, and their impact on protein function and causal links to the molecular mechanism(s) of disease expression remain unclear. Understanding the specific impact of these point mutations on PTEN function and their repercussions in cellular and tissue development is necessary for relating its disruption to disease.

As PTEN is a multifunctional protein expressed throughout the body, and both cancer and ASD are complex conditions involving diverse tissues, it is likely that VUS may disrupt distinct PTEN functions in different cell types and developmental conditions to contribute to disease. Saturation mutagenesis and massively parallel functional approaches to characterize single-nucleotide variants of PTEN have identified a large range of impacts on PTEN's lipid phosphatase activity using single-cell assays[10,11]. In order to more comprehensively assess a full range of PTEN functions, here we take a deep phenotypic profiling approach by assessing the impact of MS and nonsense (NS) variants in 18 bioassays in 5 model systems. Applying high-throughput unbiased analyses of genetic interactions of human PTEN in Saccharomyces cerevisiae we develop an assay of PTEN variant function sensitive to lipid phosphatase activity. By expressing human PTEN variants in Drosophila melanogaster we assess the rate of development in an invertebrate model, regulated by the insulin receptor pathway[12]. We test a smaller subset of PTEN variants for effects on neuronal dendritic and axonal growth, and excitatory and inhibitory synaptogenesis using rat primary neuronal cultures, processes found to be disrupted in ASD models[13] and mediated by both lipid and protein phosphatase PTEN activities[14,15]. As ASD is primarily defined as disordered behavior and sensorimotor processing, we test impact of variants on chemotaxis in Caenorhabditis elegans. In order to probe the molecular mechanisms causing variant dysfunction in these assays, we perform flow cytometry analysis of human embryonic kidney (HEK293) cells overexpressing PTEN variants to parse effects on protein stability and catalytic activity. These multi-model results provide robust validation of measures of variant function in highly diverse cellular environments to uncover complexities of the contribution of single-nucleotide mutations to protein dysfunction and pathophysiology.

## Results

### Categorization of PTEN variants.

We selected PTEN MS and nonsense (NS) mutations identified in individuals with ASD, intellectual disability (ID), developmental delay (DD), somatic cancer and PHTS, as well as variants found among the general population (Fig. 1a, Supplementary Data 1). We categorized a total of 48 variants as ASD (15 de novo) found in cases of ASD, ID, or DD. A set of 4 MS variants were classified as Somatic Cancer which have not been reported in ASD, but exhibit a high frequency of reports in the COSMIC database[16]. A further 19 variants termed PHTS were found in individuals with PHTS, but not ASD and low frequency in somatic cancer. There is high overlap in variant incidence across these disorders (Fig. 1b). We included 5 PTEN variants with well-characterized disruptions on protein function, termed Biochemical Variants. These include C124S, which is both protein- and lipid-phosphatase catalytically inactive[17]; G129E, a lipid phosphatase-dead variant[18]; Y138L, a protein phosphatase-dead variant[19]; 4A, in which the four phosphorylation sites $Ser^{380}$, $Thr^{382}$, $Thr^{383}$, and $Ser^{385}$ were replaced with alanines, creating a phospho-null construct lacking self-repression, rendering it constitutively active[20]; and C124S-4A which contains both catalytic inactive and phospho-null mutations[15]. C124S has also been found in somatic cancer[16], and G129E has been found in macrocephaly, DD[21], PHTS[22], and somatic cancer[16]. In order to understand the range of PTEN protein function existing in the general population, 8 variants were selected based on their relatively high frequency in the gnomAD database[23], termed Population Variants. Additionally, variants were included with either Predicted High Impact (17) or Predicted Low Impact (5) as determined by CADD phred version 1.0[24] or SNAP2 scores[25]. We also included 2 NS mutations, R130X and R335X, due to their common association with ASD, PHTS and somatic cancer. Many variants exist in multiple categories, and the full variants classification, reference, and CADD phred and SNAP2 scores can be found in Supplementary Data 1. We use the National Center for Biotechnology Information (NCBI) reference sequence of PTEN (NM_000314.7) as wild type (WT). PTEN variants were placed in model-specific plasmid expression constructs, and variants for yeast, Drosophila and C. elegans were expressed as non-tagged proteins, while variants for rat neurons and HEK293 were fused with a 3XHA epitope, and Superfolder Green Fluorescent Protein (sfGFP) at the N-terminus, respectively.

### Yeast gene interaction assay for lipid phosphatase function.

We took advantage of the high-throughput capacity of the Saccharomyces cerevisiae synthetic dosage lethality screen[26] as an unbiased assay to identify genetic interactions of PTEN. By overexpressing human WT PTEN (WT), empty vector (EV), or the catalytically inactive variant C124S, in ~5000 non-essential gene mutant strains of the yeast deletion collection[27], we identified 44 potential genetic interactions. Eight of which, deemed sentinels, show significant, strong negative genetic interactions with WT, but not EV or C124S, and were selected for inclusion in our assay (Fig. 1c). Overexpression of WT in all 8 sentinel strains reduced colony growth. The sentinel proteins Vac7 and Vac14 interact and promote the synthesis of $PI(3,5)P_2$ by activating Fab1, a kinase for $PI3P$[28,29] (Fig. 1d). Fig4, when recruited to the vacuolar membrane by Vac14, dephosphorylates $PI(3,5)P_2$ to form $PI3P$[29]. Therefore, lack of any of these genes alters PI3P metabolism. Likewise, the other sentinels directly or indirectly influence PI3P signaling[30]. Vam3 and Vam7 are SNARE proteins and Ypt7 is a Rab GTPase. All three are components of the vacuolar fusion complex[31–34], in which Vam7 directly binds PI3P on the vacuolar membrane to initiate fusion[35]. While yeast does not contain the primary PTEN substrate, $PIP_3$, nor an endogenous PTEN homolog, PTEN also binds PI3P, which is present in yeast[36]. We hypothesize that PTEN utilizes PI3P as a substrate in yeast and antagonizes compromised PI3P levels in these mutants.

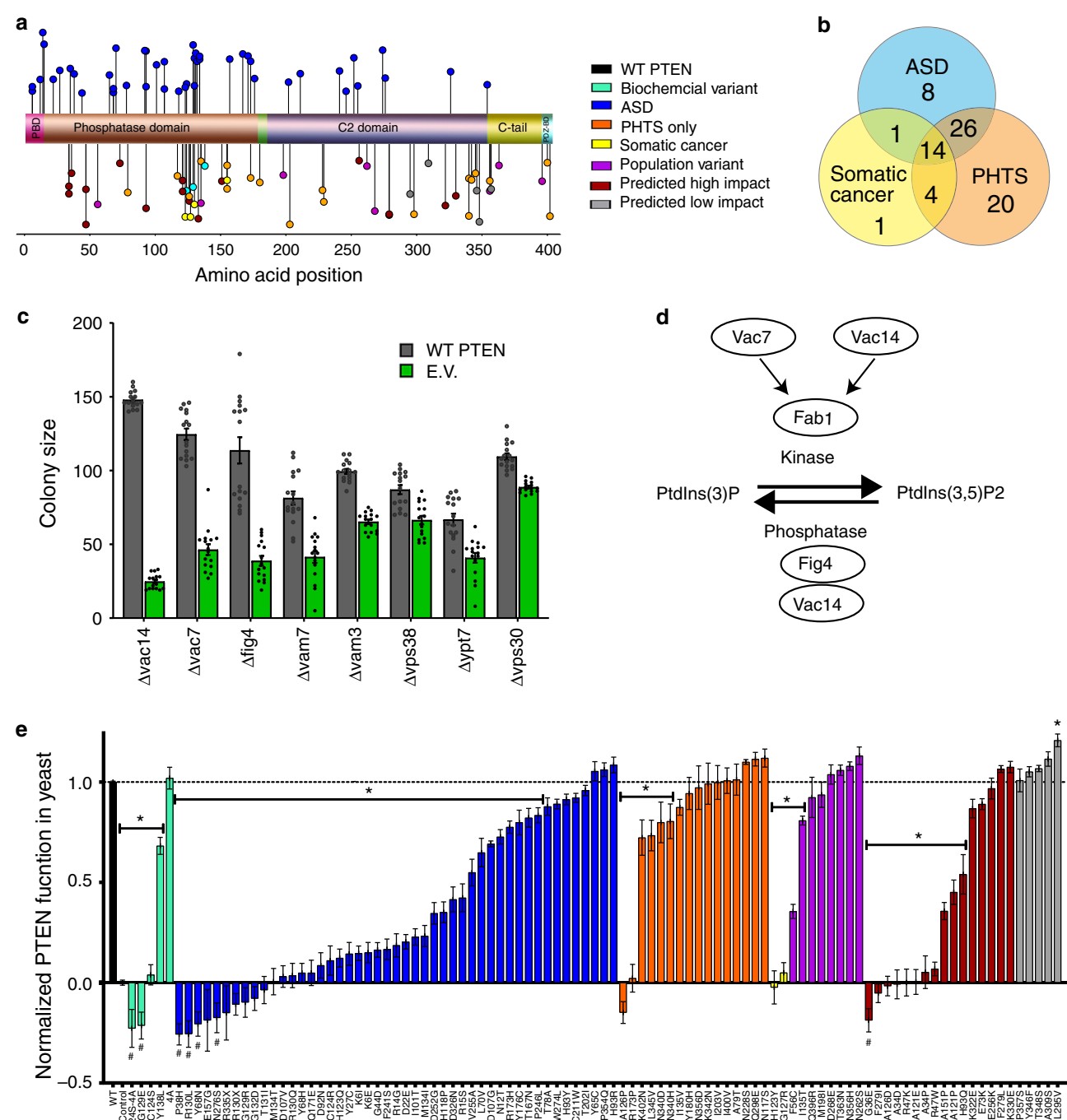

**Fig. 1 Functional assessment of *PTEN* variants for genetic interactions in yeast. a** Schematic of 106 MS and NS variants indicating their positions across the functional domains of PTEN with each variant represented by a circle colored following the coding table at right. **b** Venn diagram showing the overlap of PTEN variants used in this study identified in individuals with different disorders. The ASD category includes DD; and Somatic Cancer includes variants with 8 or more reports in COSMIC. **c** Overexpressing WT-*PTEN* in a library of 4699 yeast strains, each lacking one nonessential gene, identified 8 sentinel strains showing significant genetic interaction (five of which function in phospholipid metabolism; three in vacuole fusion). WT PTEN = gray, E.V. = empty vector, green. Data are expressed as means ± SEM (*n* = 16 for all sentinels). Two-tailed Student's *t* test for comparison between WT-PTEN and E.V. found *p* < 0.05 for all eight sentinel strains. **d** Schematic of sentinel functions in PIP3 metabolism. **e** 99 non-tagged *PTEN* variants were expressed in the sentinel yeast strain *Δvac14* and assayed for colony size. Data are expressed as means ± SEM, and variants are colored based on their categories following the same colr table in **a**. * indicates nominal *p* < 0.05 by two-tailed Satterthwaite approximation for the contrast between WT and variant in the mixed-effects model (see Methods). Individual variant Means, *n* and nominal *p*-values for all plots are provided in the Source Data file.

Next, we designed a mini-array platform to screen the genetic interactions of WT, EV, and 99 PTEN variants with each of the eight sentinel strains. As a representative data set, we present results for the yeast strain lacking Vac14 (*Δvac14*), which shows the strongest genetic interaction with PTEN and allows for the greatest dynamic range to study the effects of PTEN variants (Fig. 1e). We find that using the *Δvac14* sentinel, 37 variants, including 22 of 47 ASD-associated variants tested, exhibit complete LoF, with 6 being below LoF. In all, 27 variants retained partial function while 35 variants, including 8 ASD, retained

WT-like functionality. Similar results were found with the 7 other sentinels (Supplementary Fig. 1; Supplementary Data 2).

**PTEN variants alter developmental rate in Drosophila**. We generated 88 transgenic lines of *Drosophila melanogaster* expressing WT human PTEN and 86 PTEN variants, each integrated into the *attP2* locus[37], with *attP2* used as an empty vector (EV) control, allowing quantitative comparison of relative function between all variants by virtue of their equivalent expression levels. In *Drosophila*, reduced insulin receptor signaling slows development, causing delay in pupa formation and eclosion - the emergence of the adult fly from the pupa[12,38] (Fig. 2a). Given PTEN's function in counteracting insulin signaling, we found that overexpression of WT in all cells throughout development, led to significant delay in eclosion compared to EV, not observed in flies expressing C124S (Fig. 2b). We then screened 86 PTEN variants, including 45 ASD to compare time to eclosion against one another, EV (normalized as 0) and WT (normalized as 1) (Fig. 2c). Variants exhibited a range of functionality: with 41/86 variants being complete LoF, including 26 ASD. In total, 34/86 were partial LoF, or hypomorphic, and 11/86 variants retained WT functionality. We observed 2 variants, the PHTS N117S and Q298E, which slowed developmental rate further than WT, while the known gain-of-function (GoF) variant 4A induced lethality. Finally, we noted 6/86 variants including the lipid phosphatase-dead G129E exhibited eclosion significantly faster than EV. We find a high level of concordance (Pearson $r = 0.69$, $p < 0.0001$) between functional measures of $\Delta vac14$ genetic interactions in yeast and development in fly (Fig. 2d).

**PTEN variants impact rat neural development**. Aberrant neuronal morphology and excitatory/inhibitory synapse balance are hallmarks of ASD[39], and ASD rodent models[13]. Reducing *PTEN* expression results in increased neuronal growth, with larger soma size, increased dendritic and axonal length, and increased excitatory, but not inhibitory synapses[15,40–43]. To assay variant function in these ASD-associated growth processes, we overexpressed WT, 19 PTEN variants, or Green Fluorescent Protein (GFP) alone, in rat primary hippocampal and dorsal root ganglion (DRG) neuronal cultures. We find that overexpression of WT induced the opposite effects observed by reducing PTEN, including decreased density of PSD95 puncta, a marker of excitatory glutamatergic synapses (Fig. 3a, c), but had no impact on density of the inhibitory GABAergic synapse marker Gephyrin (Fig. 3a). WT induced a significant reduction of soma size and total dendritic arbor and axonal lengths (Fig. 3b, d–f). For PSD95 density, 14/19 variants including C124S showed complete LoF indistinguishable from GFP controls, and only 4/19 variants retained WT-like function. In addition, 11/19 variants, exhibited complete LoF activity on dendrite growth, while seven retained WT-like function. For axonogenesis, 6/19 variants were complete LoF while 13/19 were WT-like. For soma size, variants displayed a varying amount of dysfunction, with 10 variants exhibiting significant LoF. C124S and two ASD variants, I101T and G132D, increased soma size more than GFP. The Population Variant D268E appeared WT-like in all four measures of growth and synaptogenesis. Notably, the PHTS variant A79T, which is also found at increased frequency in gnomAD, exhibited a GoF phenotype in axonal growth and complete or less than LoF for PSD95 density and dendrite length.

**Variants impact C. elegans sensory processing and behavior**. Since ASD is a behaviorally diagnosed disorder, and deficits in sensory processing are a core feature of ASD present in >95% of cases[44], we tested the effects of PTEN variants on sensorimotor neural circuit function in an intact animal model. We used a

behavioral assay in *C. elegans* as they have a unique advantage of being the only animal known to survive homozygous null mutations in PTEN, allowing us to assess the functional consequences of PTEN variants in a transgenic rescue assay in vivo[45]. *C. elegans* exhibit robust navigation up a concentration gradient of NaCl. Worms harboring NS mutations in the *PTEN* homolog, *daf-18*, exhibit a complete reversal in NaCl chemotaxis due to a dysregulation in the insulin pathway acting on a single sensory neuron (Fig. 3g)[46]. Pan-neuronal expression of WT human PTEN in *daf-18* mutants was sufficient to rescue normal chemotaxis. We then used our machine vision system, the Multi-Worm Tracker[47] to automatically quantify chemotaxis in large populations of animals. We overexpressed the same WT and 19 PTEN variants tested in rat neuronal cultures in *daf-18* mutant worms and found that 14/19 variants, including C124S exhibit complete LoF, while T167N, Y176C, and D268E showed partial LoF, and A79T and P354Q retained WT-like function (Fig. 3h). The ASD variant G132D exhibited stronger negative chemotaxis than *daf-18* mutants.

**Protein instability contributes to variant dysfunction**. To identify molecular mechanisms underlying variable effects of MS variants, we measured impact on protein stability, a known mechanism of PTEN dysfunction[10,20]. We measured PTEN protein abundance for WT and 97 variants tested in yeast using western blot analysis (Fig. 4a) and found a significant reduction in PTEN levels for 31 variants compared to WT, with 21 variants having stability below 50% of WT. Notably, of 46 ASD variants measured, nearly half (22/46) had significantly reduced levels of PTEN. To examine whether the changes in protein abundance assessed in yeast is reflected in a human cellular environment, we overexpressed WT and 105 variants as sfGFP-fusion proteins in HEK293 cells, with the red fluorophore mTagRFP-T co-expressed from the same plasmid. The green-to-red ratio for each cell was quantified using flow cytometry, with a decrease from WT level indicating reduced protein stability (Fig. 4b–d). In all, 77/105 variants were significantly less stable than WT, of which 50 exhibited stability of 50% or less than WT. 19/105 showed WT-like stability, and 9/105 were significantly more stable. Of the 47 ASD variants tested, most (40/47) were significantly less stable than WT, and 29 exhibited less than 50% WT stability. Only 5, P354Q, R130L, R15S, R14G, and T78A, were not significantly different from WT. The variants H93Y and K6E were hyperstable. Interestingly, all of the Biochemical Variants exhibited significant protein instability. PTEN stability in yeast and HEK293 cells were well correlated (Fig. 4e, Pearson $r = 0.55$, $p < 0.001$), suggesting the presence of an N-terminal GFP does not affect PTEN variant stability, as shown previously[10]. Results indicate that instability of PTEN protein is a major mechanism of single-nucleotide variation-induced dysfunction.

**HEK293 AKT assay finds 25 uncharacterized dominant negative variants**. PTEN functions as a negative regulator of the PI3-AKT signaling pathway by decreasing the pool of available PI$(3,4,5)$P$_3$ via its lipid phosphatase activity, causing a reduction in the level of activated, phosphorylated AKT (pAKT)[48]. To assess the lipid phosphatase function of WT PTEN and 105 variants, we measured pAKT/AKT in insulin-treated HEK293 cells using flow cytometry (Fig. 5a-c). As expected, overexpression of WT decreases pAKT levels in an exogenous PTEN expression level-dependent manner (Fig. 5a). In all, 25/105 variants showed complete LoF and 19/105 showed partial LoF. 29/105 variants retained WT-like function, and P354Q, Q396R and the constitutively active 4A, exhibited GoF. Further, 65/105 variants had <50% functionality, including 42/47 ASD. Expression of C124S or G129E raised pAKT/AKT (Fig. 5c), consistent with a dominant

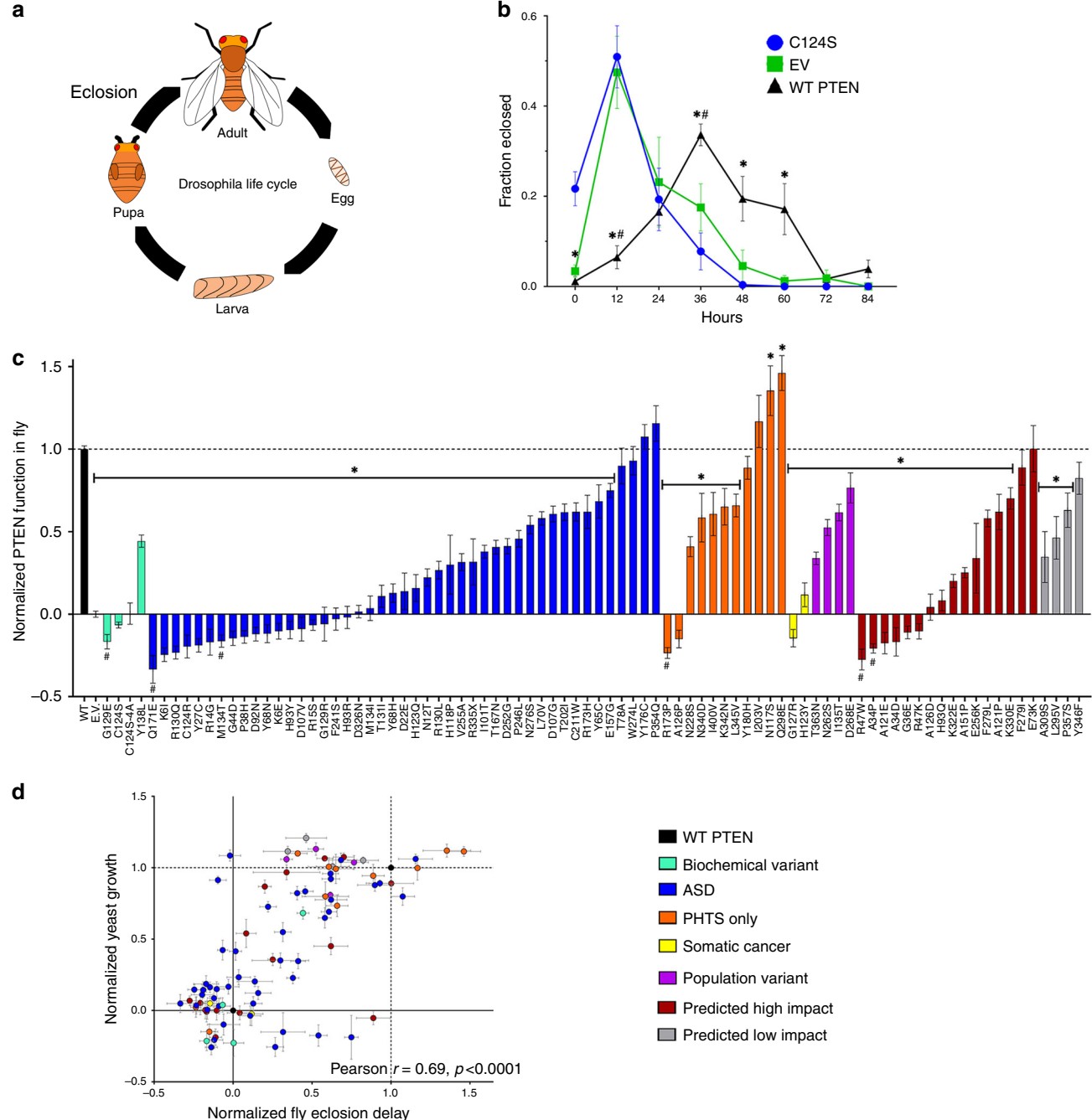

**Fig. 2 Functional assessment of *PTEN* variants in *Drosophila* development. a** Schematic of *Drosophila* life cycle highlighting eclosion transition form pupa to adult. **b** Transgenic *Drosophila* overexpressing WT-PTEN (black triangles) exhibited delayed development, indicated by increased time to reach eclosion compared to empty vector (EV, green squares), and the catalytically inactive variant C124S (blue circles). Data are expressed as means ± SEM. *$p < 0.05$ for WT vs. C124S, #$p < 0.05$ for WT vs. EV by two-tailed Student's *t* test. **c** Effect on time to eclosion for 88 strains of transgenic flies each expressing a different non-tagged *PTEN* variant, with variants colored by category following the color table below. Data are expressed as means ± SEM. *$p < 0.05$ for the contrast between WT and variant, and #$p < 0.05$ for the contrast between EV and variant by two-tailed Satterthwaite approximation in the mixed-effects model (Methods section). **d** Correlation of genetic interaction values from yeast sentinel Δ*vac14* and developmental delay in *Drosophila* for 84 PTEN variants. Variants are colored following the same color table used in **c**. Data are expressed as means ± SEM, Pearson $r = 0.69$, $p < 0.0001$. Individual variant means, error, *n* and nominal *p*-values for all plots are provided in the Source Data file.

negative phenotype, as reported previously[10,14,49,50]. Remarkably, we identify 27 additional variants (25 uncharacterized) exhibiting dominant negative activity, including 17 ASD. To further investigate these phenotypes, we used CRISPR/Cas9 to create a bi-allelic knockout (KO) of PTEN (PTEN-KO) in HEK293 cells and repeated the pAKT/AKT assay for 32 variants. For 21 of 28 variants exhibiting dominant negative activity in parental cells,

the absence of endogenous PTEN abrogated this effect (Fig. 5d). Interestingly, the GoF of constitutively active 4A was markedly reduced in PTEN-KO, yet still significantly more active than WT, suggesting it may enhance endogenous PTEN activity when present. We found only 6/32 variants tested (A126P, H123Q, P38H, R130Q, A126D, and R130L) exhibiting >10% change in stability between parental and PTEN-KO cells suggesting that

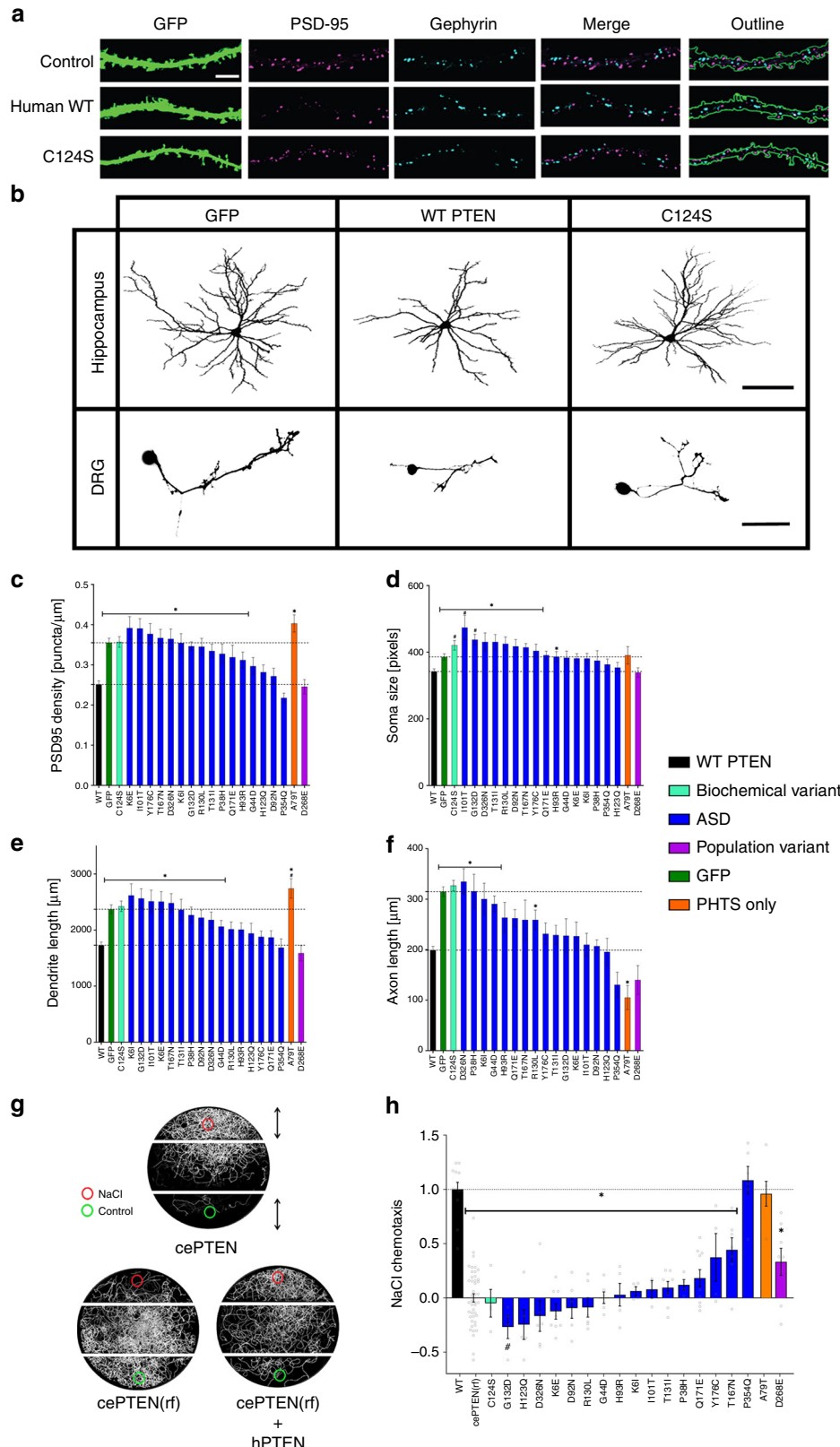

interactions with endogenous PTEN had minimal influence on variant stability (Supplementary Fig. 2).

**Varied functional correlations based on stability effects.** Overall, we find a range of strength of correlation between assays and

model systems (Supplementary Fig. 3), with strongest correlations observed between the 8 yeast sentinels, and between the HEK293 pAKT/AKT assay and yeast genetic interactions (Fig. 5e; Pearson $r = 0.79$, $p < 0.0001$ for $\Delta vac14$), fly development (Fig. 5f; Pearson $r = 0.78$, $p < 0.0001$), rat axonal outgrowth (Pearson $r = 0.73$, $p < 0.001$), and worm sensorimotor behavior (Pearson $r = 0.82$,

**Fig. 3 *PTEN* variant impact on rat neural growth and sensorimotor behavior in *C. elegans*. a** Representative immunofluorescence images of proximal dendrites from primary cultured hippocampal pyramidal neurons co-stained for the postsynaptic markers of excitatory (PSD-95) and inhibitory (Gephyrin) synapses. Similar results were seen in 128 replicates of WT PTEN, 109 replicates for GFP Control, and 75 replicates for C124S. Scale bar = 5 μm. **b** Representative images of cultured hippocampal and DRG neurons expressing GFP alone, or with WT-*PTEN* and *PTEN*-C124S (scale bar = 100 μm for Hippocampus, and 60 μm for DRG). Similar results were seen in replicates for WT PTEN (132 hippocampus, 463 DRG), GFP Control (125 hippocampus, 488 DRG), and C124S (84 hippocampus, 444 DRG). Effect of overexpression of 20 3XHA-tagged *PTEN* variants on excitatory synapses (**c**), soma size (**d**), total dendritic arbor length (**e**) in hippocampal neurons, and axonal length in DRG neurons (**f**). Data are expressed as means ± SEM. *$p < 0.05$ by two-tailed Satterthwaite approximation as in Fig. 1 (Methods section). **g** Wild type *C. elegans* (cePTEN) exhibit chemotaxis towards a NaCl source, while worms harboring a NS mutation in the worm homolog *daf18* (cePTEN(rf)) exhibit chemotaxis deficits, which is rescued by overexpression of human PTEN (cePTEN(rf)+hPTEN). **h** Impact of 20 non-tagged *PTEN* variants on NaCl chemotaxis in *C. elegans* is shown by scoring salt preference as (A–B)/(A+B). Data are expressed as means ± SEM and are normalized to cePTEN(rf) + hPTEN = 1 and cePTEN(rf) = 0. Variant data represented in all histograms are color-coded by variant category depicted in the color table. *$p < 0.05$ compared to WT, #$p < 0.05$ compared to cePTEN(rf) by two-tailed Satterthwaite approximation. Individual variant means, error, *n* and nominal *p*-values for all plots are provided in the Source Data file.

$p < 0.0001$). Strong correlations between assays employing non-tagged PTEN (yeast, *Drosophila* and *C. elegans*), 3XHA-tags (rat) or sfGFP-fusions (HEK293), suggest minimal impact of N-terminal exogenous motifs on PTEN variant function.

In order to identify features of variant impact contributing to varied phenotypes in different assays we investigated the relationship between protein function and stability. To determine whether function directly tracks stability we subtracted the normalized stability (nS) of each variant from its normalized function value (nF) in each assay. If function directly follows stability then nF-nS≈0, while values closer to 1 or −1 would suggest function/dysfunction independent of stability. Positive values indicate variants with greater function than predicted from stability, while negative values indicate variants with less function than predicted from stability. We used measures of protein abundance in yeast for nS for yeast, and stability measures in HEK293 for all other assays. Interestingly, we find domains within PTEN in which variants predominantly exhibited either stability-independent function (nF-nS<>0), or stability-dependent function (nF-nS≈0; Fig. 6a, b). Stability-independent domains, encompassing variants which largely exhibit greater dysfunction than predicted from their instability (nF-nS ≈ −1), are associated with previously well-characterized substrate binding and catalytic domains including within the N-terminal phosphatase and the PIP3-binding domain (PBD) (amino acid (AA) positions 1–55; the WPD-loop (AA 92–93); and the P-loop (AA 123–131)[51]. Therefore, we define stability-independent domains as AA 1–55, 92–93, and 123–131, and variants outside these regions, largely exhibiting nF-nS≈0, as being in stability-dependent domains (Fig. 6a, b). Variants within these two groups showed distinct distributions when plotting the frequency of nF-nS values (Fig. 6c). Plotting the HEK293 pAKT/AKT function against stability reveals that for variants in stability-dependent domains, function tracks stability, yielding a range of function from WT to partial and full LoF (Fig. 6d). For these variants, instability is likely the primary molecular mechanism underlying dysfunction. In contrast, variants within stability-independent domains exhibit complete LoF or dominant negativity irrespective of level of instability, likely indicating molecular mechanisms of dysfunction associated with aberrant substrate binding or catalytic activity. We find similar distributions of nF-nS values using data from yeast or combining results from all models (Supplementary Fig. 4), and when removing variants with function and stability values not significantly different from WT (Supplementary Fig. 4f).

Repeating correlation analyses for variant impact between assays produces dramatically different results when comparing variants within stability-dependent and -independent domains (Fig. 6e, f; Supplementary Fig. 5a, b). Correlations were high between all assays for variants within stability-dependent

domains, indicating that mechanisms of protein instability are independent of model system. Correlations for variants in stability-independent domains, however, were poor across models, suggesting that different assay are selectively sensitive to dysfunction of different substrate binding and catalytic domains.

**Relationship of PTEN variant dysfunction and disease**. Attempts to associate MS induced dysfunction to specific disease states is a major role for functional variomics for PTEN, however, this goal is challenged by variants selection bias and high incidence of the same variants in multiple disorders. To investigate this relationship, we reclassify variants as ASD or PHTS if they have been identified in any individual with these disorders (including ID or DD in ASD), and as Somatic Cancer if variants have eight or more cases reported in COSMIC. For this analysis, we allow the same variants to be in multiple classes. Of the 106 PTEN variants we tested, 49 have been identified in individuals with ASD, DD, or ID, with 40/49 also found in PHTS, 15/49 in somatic cancer, and 14/49 in all three disorders (Fig. 1b; Supplementary Data 1). Further, 64 of the total 106 are PHTS, 20 are Somatic Cancer, and 18 found in both disorders. We find that in yeast and HEK293 lipid phosphatase assays, ASD, PHTS, and Somatic Cancer variants all performed significantly worse than Population Variants. However, no significant difference was found for variant function between these disorders, except between PHTS and Somatic Cancer in the HEK293 pAKT/AKT assay (Supplementary Fig. 6a), but this difference was lost when variants we determined to be likely benign due to WT stability and function in multiple models (below) were removed from analysis (Supplementary Fig. 6b). We note that this analysis relies on assumptions of disease causality that are uncertain, so we interpret the negative result as inconclusive on whether there are disease-specific differences in PTEN variant effects. In contrast, we find that variant impact on function in the yeast, fly and HEK293 assays all show significant predictive powers when stratified by their frequency of occurrence in somatic cancers as reported in COSMIC (Fig. 7).

**High-confidence variant classification**. We classified 106 PTEN variants according to their impacts on eight functional assays encompassing a range of phenotypes between dominant negative to GoF, and 2 abundance/stability assays (Fig. 8a). Next, we classified variants following a binary LoF/WT-classifications with a cut-off for LoF at <50% of WT function[52] (Fig. 8b). For each variant, a score from 0 to 1 is then obtained by dividing the number of LoF assays by the number of assays performed, and then classified as follows: Likely Benign variants have scores <0.25 and are WT-like in at least two assays; Likely Pathogenic have scores >0.5 and are

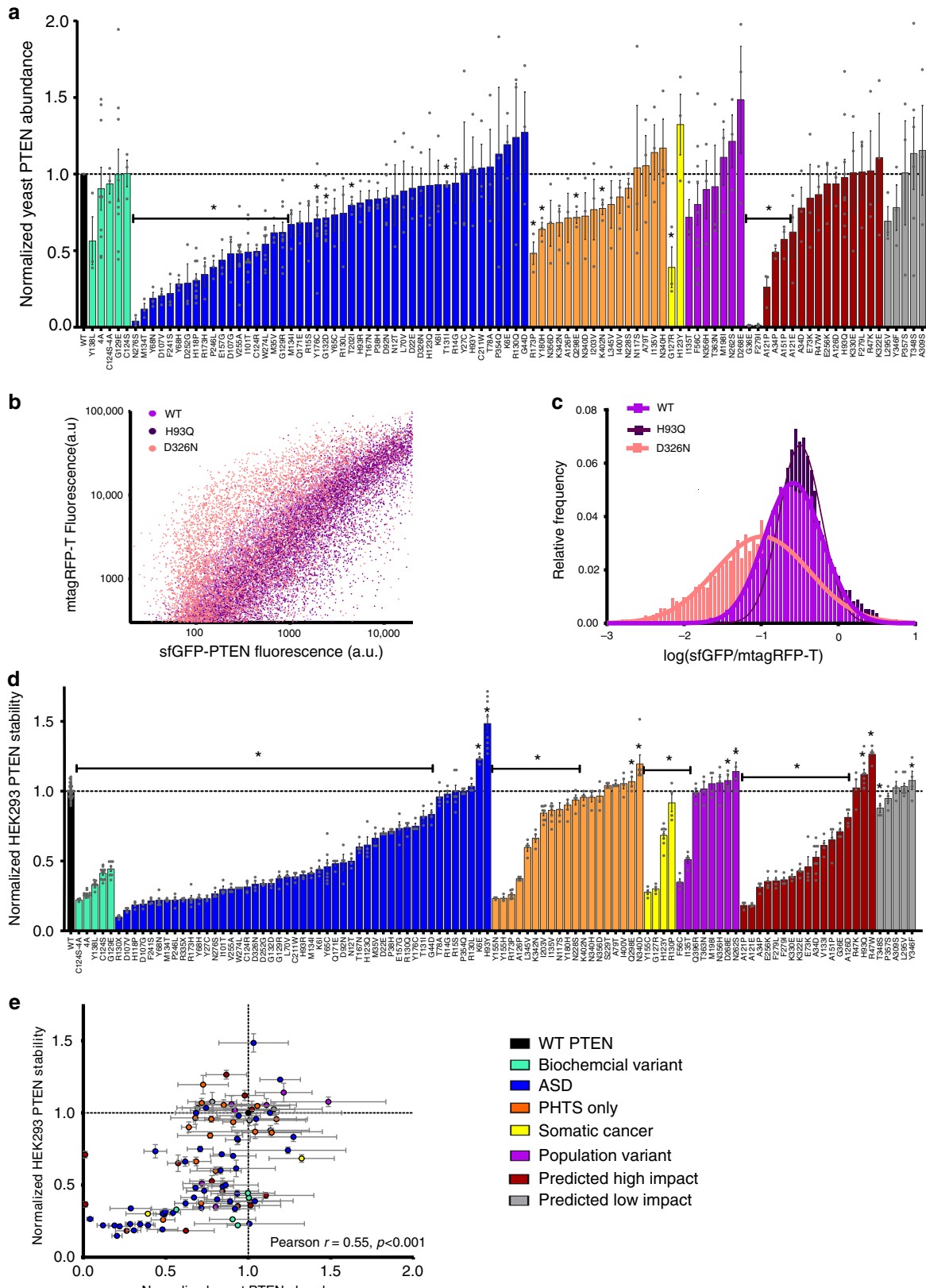

LoF in at least two assays; Pathogenic have scores ≥0.75 and are LoF in at least three assays; and VUS have scores ≥0.25, but <0.5 or too little assay data yielding conflicting results. Stability was only included in discerning variant profiling for unstable proteins, since WT-like stability is not predictive of WT-like function, but instability correlates with dysfunction.

Of the 106 variants tested, we classify 50 as Pathogenic, including 31 ASD. We further classify 10 variants, including 4 ASD, as Likely Pathogenic. We consider 24 variants to be Likely Benign, including 3 ASD: P354Q, T202I and T78A; and 12 PHTS: I135V, N340H, N356D, K402N, N117S, Y180H, I203V, Q298E, N340D, K342N, L345V, and I400V. We are unable to make a

**Fig. 4 PTEN variants destabilize PTEN. a** The abundance of 97 PTEN variants overexpressed in yeast was assayed by western blot. Variants are color-coded by category depicted in the color table at the bottom right. Quantified values by band densitometry were normalized to WT-PTEN = 1 and empty vector = 0. Data is expressed as mean ± SEM. *$p < 0.05$ compared to WT by two-tailed Satterthwaite approximation (Methods section). **b** Representative scatter plot of single-cell fluorescence intensities from flow cytometry for overexpressed sfGFP- fusions of PTEN WT (light purple), D326N (pink), and H93Q (dark purple) versus their level of transfection as visualized with mTagRFP-t. **c** Histogram showing the relative frequency of sfGFP/mTagRFP-t ratio for the same variants in **b**. **d** Protein stability of 105 PTEN variants assayed in HEK293 cells by flow cytometry calculated as median of sfGFP/mTagRFPt and expressed as normalized to WT = 1. Data is expressed as mean of well replicates ±SEM, and variants are color-coded by category depicted in the color table at the bottom right. *$p < 0.05$ compared to WT by two-tailed Student's *t* test. **e** Plot of yeast abundance and HEK293 PTEN stability variant data sets with color coding as in **a** and **d**. Data are expressed as means ± SEM, Pearson $r = 0.55$, $p < 0.0001$. Individual variant means, error, *n* and nominal *p*-values for all plots are provided in the Source Data file.

functional classification for 22 variants due to either insufficient data or conflicting results between assays and thus retain the classification of VUS. These include the ASD variants M35V, L70V, H123Q, E157G, C211W, W274L, Y176C, and Y65C, as well as the PHTS variants A79T, N228S, and S229T.

## Discussion

We have taken a deep phenotyping approach to comprehensively assess the impact of single amino acid changes on PTEN function. We use five phylogenetically diverse model systems and phenotypes spanning molecular function to behavior in order to measure variant impact on the multiple complex roles of this protein. By taking advantage of the high-throughput nature of yeast, fly, and HEK293 models, we are able to conduct in-depth analyses of ~100 variants, allowing comparison of effects of overexpression in highly diverse cellular environments. These high-throughput assays allowed selection of a smaller subset of variants exhibiting a range of dysfunction phenotypes to test in lower-throughput neuronal and behavioral assays with more direct relevance to ASD pathophysiology. By utilizing a multi-model system approach including measures of protein stability and function, we identify a diversity of variant impact caused by distinct molecular mechanisms, which would not be apparent from single-model approaches.

As previously reported[10], we find that PTEN variants commonly decrease protein stability. We find that MS variants induce a full range of instability, which was largely consistent between yeast and HEK293 models. Further, for ~50% of PTEN variants tested, located throughout the protein structure, there was strong correlation between instability and dysfunction, resulting in a corresponding range of partial to complete LoF. The consistency of this correlation across models suggests conserved mechanisms for MS-mediated PTEN degradation. For these variants, reduced protein abundance is likely the molecular mechanism underlying dysfunction in each assay, and if consistent under physiologic expression in humans, similar levels of haploinsufficiency may contribute to disease expression.

For other variants, including some exhibiting instability, levels of stability did not correlate with protein function, indicating additional molecular mechanisms underlying dysfunction. 16 variants, including 4 A, exhibited more function than expected from their stability, suggesting GoF with or without instability. However, most of these variants exhibited greater dysfunction than expected from their instability. These variants largely localized to well-characterized substrate binding and catalytic domains, including the PIP3-binding and N-terminal region of the phosphatase domain, and the catalytic pocket encompassing the WPD- and P-loops. Phenotypes were typically complete LoF or dominant negative/greater than LoF, rather than partial LoF, indicating that these functional domains are intolerant to modification.

The large number of variants exhibiting dominant negativity in HEK293 cells was unexpected. While C124S and G129E have

previously been established as dominant negatives, such a characterization of other PTEN variants is sparse[10,14,49,50]. Remarkably, we find 29 variants, including 17 ASD, showing dominant negativity in measures of activated AKT. Furthermore, P38S and R130G, two different amino acid substitutions at the same positions as MS variants exhibiting dominant negativity here (P38H, R130L, R130P, and R130Q), have previously been reported to have a dominant negative effect on pAKT levels[10,49]. We also find 6 dominant negative variants not found in ASD, but identified in PHTS and/or somatic cancer. The mechanism underlying PTEN variant dominant negative activity remains unclear, but potential mechanisms include substrate sequestration[53], dimerization-mediated inhibition of WT[53], or substrate switching[54]. It is challenging to predict the effects on pAKT levels by substrate sequestration from PTEN variants lacking phosphatase activity. Our data, however, fits a simpler model in which dominant negativity if effected through heterodimerization of dysfunctional PTEN variants with WT, resulting in inhibition of WT function[49]. Both substrate sequestration and dimerization-mediated inhibition models are supported by our finding that dominant negativity was lost for most variants in the absence of WT in our PTEN KO cell line. Alternatively, it has been proposed that the variant A126G within the P-loop substrate binding site induced a shift in affinity for different substrates, converting PTEN from a 3- to 5-phosphatase, resulting in AKT activation rather than inhibition[54]. While substrate switching would not be consistent with variants which lose dominant negativity in PTEN KO cells, it might explain the few variants, including A126D, which retain this behavior. Identifying dominant negative variants is of significant clinical value since they may produce outcomes distinct from LoF variants producing haploinsufficiency. Whether phenotypes identified in an overexpression assay are conserved at physiological levels, as shown for C124S and G129E[49], requires further study. Overall, these results suggest multiple mechanisms of MS mutation-induced PTEN dysfunction: instability producing haploinsufficiency, and loss of substrate binding or catalytic function leading to either complete LoF or dominant negativity.

We found varied strength of correlations between functional measures of PTEN variants across assays and model systems overall, but high correlations for variants in which dysfunction is due to instability. This suggests conserved mechanisms of MS-mediated protein degradation, resulting in reduced protein affecting all assays similarly. Increased intra-assay and -model variability was observed for variants whose dysfunction was associated with mutations in substrate binding and catalytic domains, likely reflecting selective sensitivity of each assay to distinct functions of PTEN. For example, for all eight sentinel assays in yeast, variants within the WPD were better tolerated than in other model systems, potentially due to differences in steric requirements for interactions with the substrate PI3P in yeast and the primary substrate PIP3 in other models. We also find that the sentinels Δvac14, Δfig4, and Δvac7, which mediate PI3P-to-PI(3,5)P$_2$ interconversion, are particularly sensitive to

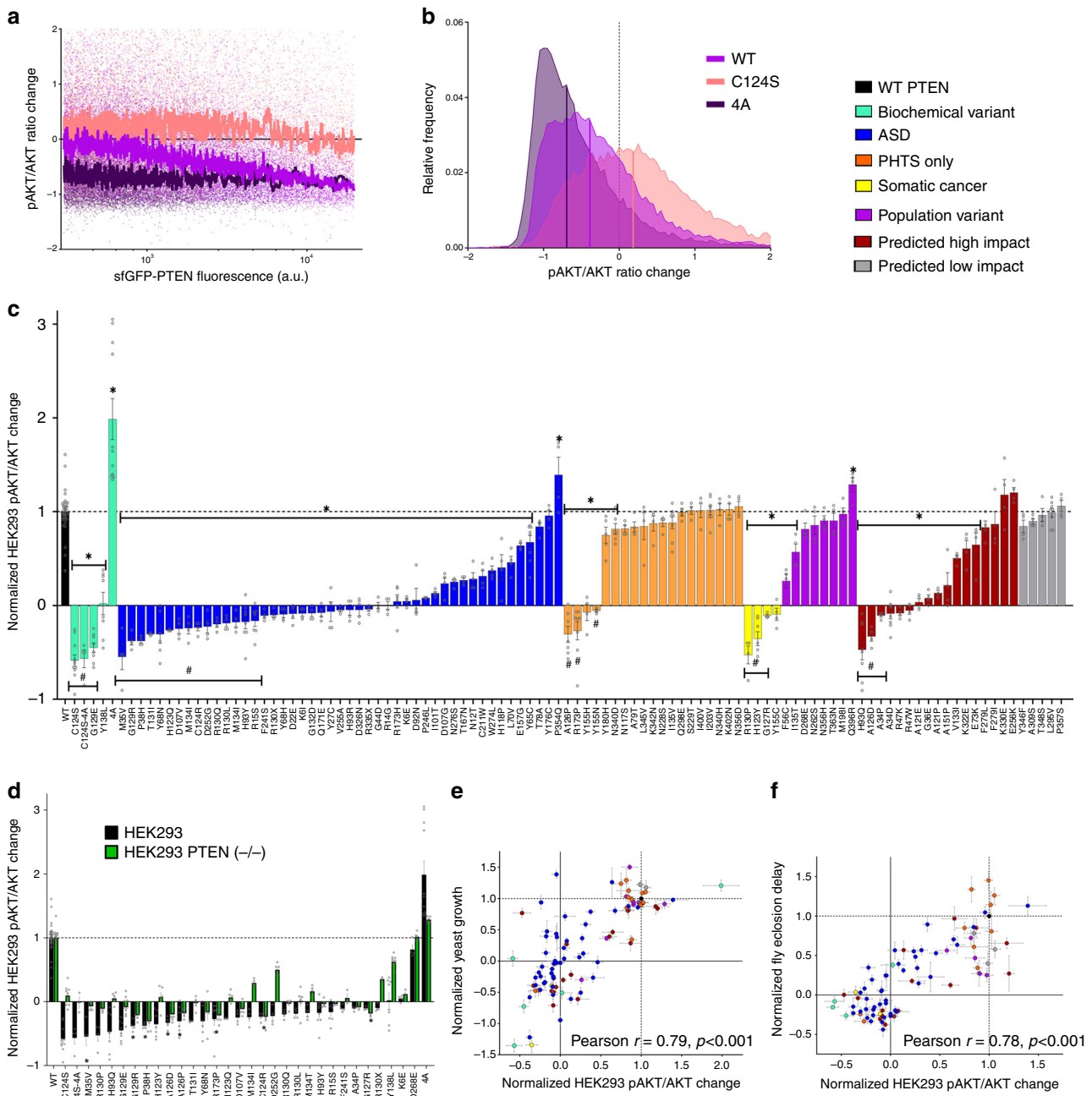

**Fig. 5 PTEN variant impact on pAKT/AKT reveals LoF and dominant negative activity. a, b** Flow cytometry single-cell rolling median of pAKT/AKT ratios versus expression levels of sfGFP-PTEN in HEK293 cells for WT (light purple), C124S (pink) and 4A (dark purple) variants and the corresponding frequency histogram in **b**. Vertical bars indicate median values. **c** Functional impact of 105 sfGFP-tagged PTEN MS variants on pAKT/AKT levels. Data plotted are median differences between transfected and in-well untransfected cells for each variant, with variants color-coded by category as depicted in the color table at top right. Data are expressed as mean of well replicates ±SEM. Values are normalized to WT = 1 and no difference to untransfected = 0. *$p < 0.05$ compared to WT, #$p < 0.05$ compared to 0 by two-tailed Student's $t$ test. **d** Relative pAKT/AKT changes are shown for variants exhibiting dominant negative effects on pAKT/AKT, as well as the constitutively active 4A in both parental (black) and a PTEN-KO (green) HEK293 cell line. Data are expressed as mean of well replicates ±SEM. Values are normalized to WT = 1 and no difference to untransfected = 0. *$p < 0.05$ comparing each variant in parental and a PTEN-KO HEK293 cell line by two-tailed Student's $t$ test. **e, f** Plots of PTEN variant function by pAKT/AKT changes in HEK293 cells versus variant function of genetic interaction with Δ$Vac14$ in yeast (Pearson $r = 0.79$, $p < 0.0001$), and rate of eclosion in *Drosophila* (Pearson $r = 0.78$, $p < 0.0001$). Data are expressed as means ± SEM with variants color-coded as in **c**. Individual variant means, error, $n$ and nominal $p$-values for all plots are provided in the Source Data file.

mutations in the PIP3-binding domain, implicating PTEN-mediated lipid interconversion in these assays. The other 5 sentinels, Δ*vam3*, Δ*vam7*, Δ*vps30*, Δ*vps38*, and Δ*ypt7*, are more sensitive to variants located adjacent to the CBR3 loop which mediates PTEN binding to endosomal membrane PI3P[36],

potentially implicating sensitivity to PTEN's role in endosomal trafficking. The fly eclosion assay, however, exhibited heightened sensitivity to mutations in the TI-loop. Other functional domains exhibit conserved sensitivity across models, including the beginning of the phosphatase domain, bordered by variants D22E to

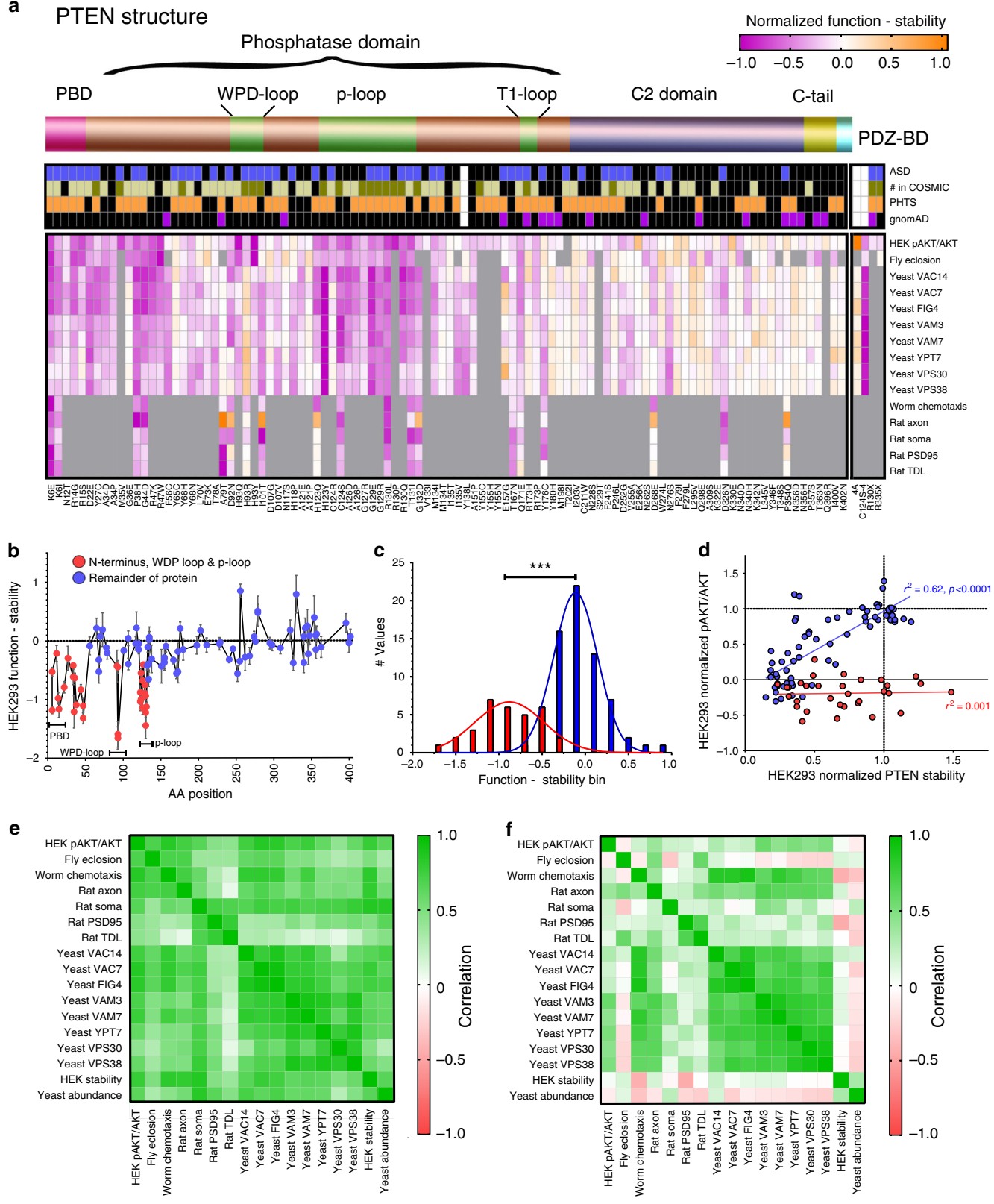

R47K and R47W, the PIP3-domain and the P-loop. Results from all assays we employed found that variants in the C-terminal region encompassing the C2 domain, serine/threonine-rich C-tail and PDZ domain largely exhibited stability-dependent function. Given that phosphorylation of C2 and the C-tail regulate its membrane association[55], and variants within the C2 domain

exhibit LoF due to reduced nuclear localization[14], it is likely that additional assays sensitive to these PTEN functions would identify additional stability-independent regions within the C-terminus. Altogether, our results highlight the strength of a multiple-assay and -model system approach for fully assessing variant impact on multifunctional proteins.

**Fig. 6 Functional and stability data identifies distinct mechanisms of molecular dysfunction. a** A normalized plot displaying stability values subtracted from function values for each assay with variants displayed according to their amino acid position below a schematic of PTEN structural domains. Normalized function – normalized stability scores are depicted as a heat map in which a score of 0 (white) indicates variants whose function matches their stability. A positive score (orange) indicates higher function than predicted from instability, while negative scores (magenta) indicates greater dysfunctional than instability (color scale at top right). **b** Variant function-stability scores for pAKT/AKT assay in HEK293 cells plotted against amino acid position with variants separated into N-terminus, WPD-loop and P-loop domains (red), and variants outside these domains (blue). Data are expressed as means ± SEM. **c** Frequency distribution of normalized function - stability scores for pAKT/AKT assay in HEK293 assay showing two distinct populations, with red and blue bars denote variants as in **b**. ***$p < 0.0001$ by two-tailed Student's $t$ test. **d** HEK293 pAKT/AKT assay data plotting PTEN variant function vs. stability with data separated by variants as in **b**: functional dysfunction of variants that is explained by stability (blue), and not (red). **e** High correlations are found for variant impact on multiple assays for variants in which dysfunction is associated with instability (blue variants in **b**–**d**). Color scale at right. **f** Weaker cross-assay correlations are seen for variants in catalytic domains, which exhibit greater dysfunction then explained by instability (red variants in **b**–**d**). Color scale at right. Individual variant means, error, $n$ and nominal $p$-values for plot are provided in the Source Data file.

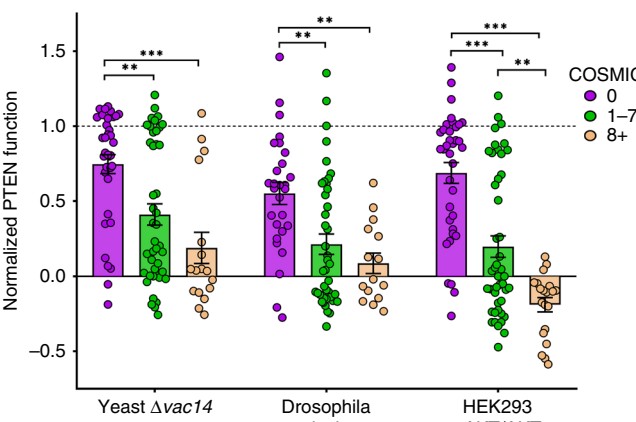

**Fig. 7 Severity of variant dysfunction correlate with occurrence in cancer.** Distribution of normalized PTEN variant function by its frequency within the COSMIC database, with variants with a frequency of 0 in purple, 1–7 reports in green, and 8 or more reports in tan. Mean PTEN function is plotted as bars ±SEM. *$p < 0.05$, **$p < 0.005$, ***$p < 0.0005$ by two-way ANOVA. Individual variant means, error, $n$ and nominal $p$-values for plot are provided in the Source Data file.

Complex patterns of variant effects on our measures of neuronal morphogenesis and sensorimotor behavior suggests PTEN has multiple functions in neurons. This is supported by previous findings that both lipid and protein phosphatase PTEN activity participate in synaptogenesis[15,56]. Here, we find similar impact on excitatory synaptogenesis and dendritogenesis, indicating common PTEN action, which may be due to synapse formation directly affecting dendritic growth[57]. Such an interaction would not occur in the axonogenesis assay, since axon growth is measured at stages prior to onset of synaptogenesis. For some variants, results from rat neuronal growth and worm behavior assays differed from yeast and fly assay. For example, H123Q has less impact on the neuronal and behavior assays, while greater impact was seen for a number of ASD variants, including D326N, T167N, I101T, as well as the PHTS variant A79T.

A critical role of functional variomics is establishing whether VUS encode dysfunctional proteins potentially causal to disease, or protein with WT function and likely benign. Given the overlap of specific variants associated with ASD, PHTS, and somatic cancer, limited annotation of patient phenotypes, and our inclusion of variants identified in single cases of ASD and PHTS, we could not attribute distinct molecular mechanisms of PTEN dysfunction to distinct disease states. However, of the 74 variants we tested found in ASD, PHTS, and/or somatic cancer, we find 10 with phenotypes not significantly different from WT, or with >50% WT function in all assays tested. We predict that these

variants are likely benign and not contributing to disease. For ASD, T78A, T202I, and P354Q, show WT activity across assays, and P354Q is the third-most common variant identified in the population (gnomAD), supporting classification as benign. The remaining ASD variants exhibit dysfunction ranging from partial to complete LoF, and dominant negativity. Furthermore, we find a strong relationship between variant impact on multiple assays and their frequency of detection in cancerous tumors. All of the 19 MS variants we examined with at least eight reports in somatic tumors showed LoF in all high-throughput models, particularly those located proximal to the lipid phosphatase domain. Altogether, our results strongly support a multi-model system approach for high-confidence profiling of genetic variants.

## Methods

**PTEN Variant selection and annotation**. All assays were performed using the full-coding sequence for the most abundant PTEN isoform (RefSeq NP_000305.3, Uniprot P60484, GenBank AAB66902.1). MS and NS base substitution variants were selected from VariCarta[58], SFARI Gene[59], ClinVar[60], COSMIC[16], and ExAC[61] databases and the literature. Site directed mutagenesis was used to generate these variants. We used VariCarta to harmonize and annotate the ASD-associated variants. Biochemical Variants were selected from the literature. Population Variants were selected which have relatively high allele frequency in ExAC and/or gno-mAD[23] and the absence of disease-association with CADD < 25. Predicted High Impact or Predicted Low Impact variants were selected based on the CADD phred version 1.0 31 or SNAP2 scores. SNAP2 scores were obtained by correspondence with the authors[25]. Variants were obtained in a Human Genome Variation Society (HGVS) protein mutation format and back-converted to genomic coordinates using TransVar[62], using Reverse Annotation: Protein mode, GRCH37/hg19 as the reference genome and RefSeq for the annotation database. Resulting genomic coordinates were then annotated using both a local Annovar instance[63] and the wAnnovar web server (ran on 24 June 2019) for additional annotations. Variants were generated using site-directed mutagenesis using Agilent Pfu Polymerase and move to expression vectors either using Gateway LR Clonase (Invitrogen) or restriction cloning. Primers used for each variant are listed in Supplementary Data 3.

**Gene interaction screen in Saccharomyces cerevisiae**. For all yeast assays, we expressed non-tagged PTEN variant proteins. For the synthetic dosage lethality (SDL) screen. the Y7093 strain was transformed with the pGAL1/PTEN plasmid mated to the haploid deletion mutant array at a density of 1536 spots per plate using a Singer RoToR HDA robot (Singer Instruments, Somerset, UK)[64,65]. The resulting diploids were copied in triplicate onto enriched sporulation medium and incubated at 25 °C for 14 days. MATa haploid cells were generated by germination on SC-His/Arg/Lys + Canavanine/Thiamine. Triple mutants were selected by two rounds of incubation on SC-His/Arg/Lys/Ura + Canavanine/Thiamine/G418 medium. A control set of single mutants was generated by two rounds of incubation on SC-His/Arg/Lys + Can/Thia/G418 + 5-Fluoroorotic acid. Colony size was determined under PTEN-inducing conditions with 2% galactose and 2% rafinose and ratios determined by comparing colony size on control (+Ura) versus experimental (−Ura) plates using the analysis software Balony[65].

**Mini-Array analysis in Saccharomyces cerevisiae**. A sentinel deletion plate was created with the eight sentinels such that each sentinel was represented in 4 × 4 colonies 12 times on 1536 dense plates. This was mated with a set of master query plates which each contained seven variants and pEGH with rows of PTEN in an alternating pattern such that each variant or pEGH spot was paired with a PTEN spot directly beneath. In similarity with the SDL screens, the query and sentinel plates were mated then underwent sporulation under enriched sporulation

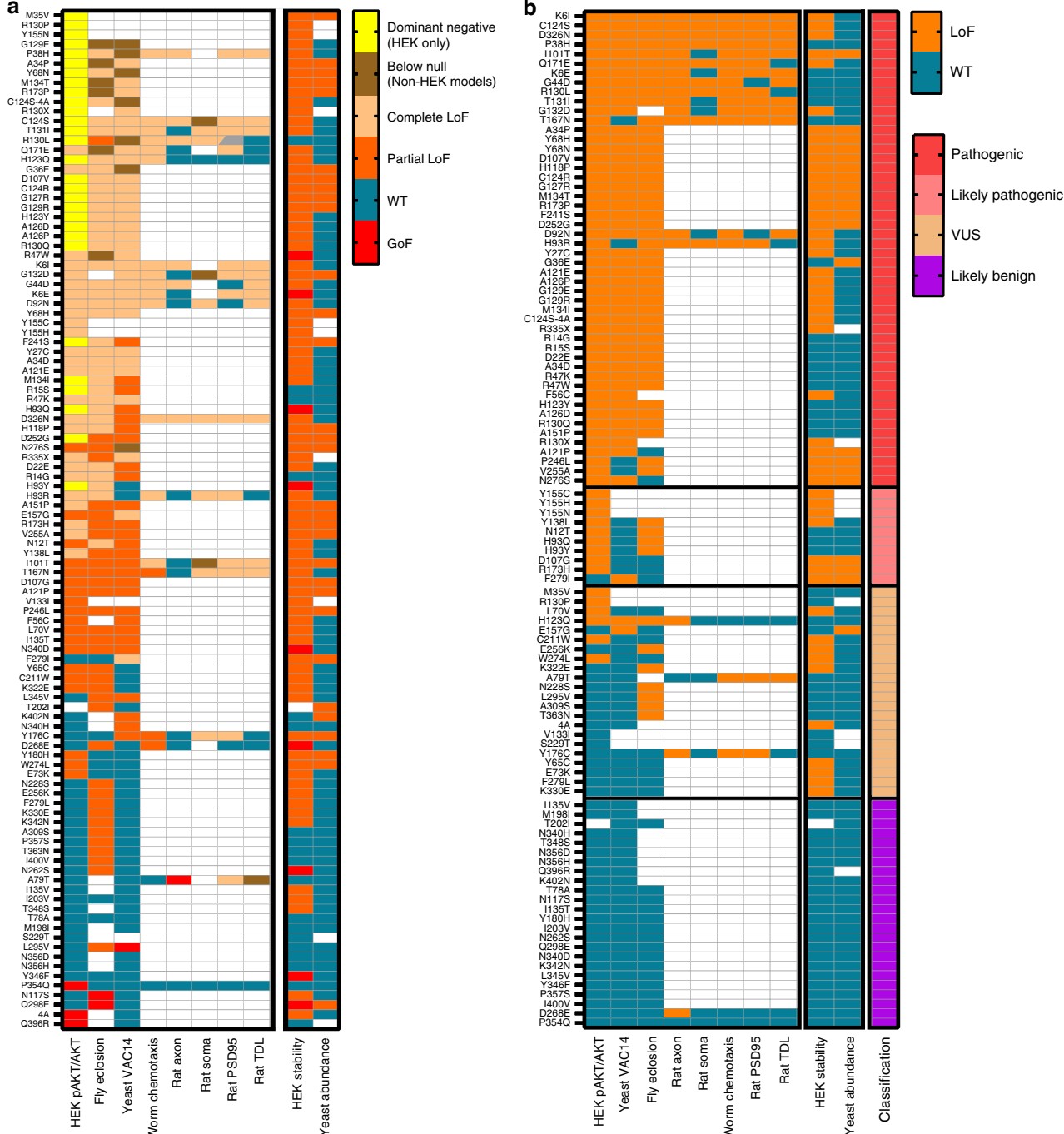

**Fig. 8 PTEN variant classification based on 9 assays of variant functions. a** The results of 9 functional assays for 106 PTEN variants are summarized according to the following criteria: WT-like variants exhibit no differences to WT, GoF exhibit function values significantly greater than WT, Partial LoF exhibit function significantly greater than null but significantly less than WT, Complete LoF exhibit no differences to null, dominant negative exhibit function significantly less than both null and WT (Below LoF for fly and yeast assays). Classification are based on statistically significant differences of at least nominal $p < 0.05$ compared to the WT-PTEN, null or both using the statistical method of each individual assay. White cells indicate that the variant was not tested in that assay. Color-coding defined in color table at right. **b** PTEN variant predicted impact based on a binary notation of either LoF (<50% of WT effect) or WT-like (≥50% of WT effect) and classified accordingly based on their frequency of LoF (pathogenic) or WT (likely benign) in 9 assays. Color-coding defined in color table at right. Source data are provided as a Source Data file.

conditions at 25 °C for 14 days. Haploid cells expressing the query plasmids in the deletion strains were selected for as in the traditional SDL and colony size was scored using Balony software after induction of gene expression using 2% galactose and 2% raffinose in the media.

**Developmental rate assay in transgenic Drosophila.** Human PTEN variants were transferred into the pGW-HA.attB destination vector[66] by Gateway-mediated

recombination. Expression of non-tagged PTEN variants was driven by a 5xUAS-hsp70 promoter. These were integrated into the *attP2* locus of the *Drosophila* genome, by phiC31-integrase (performed by Rainbow Transgenics Inc, CA) which inserts a single copy of the gene and ensures reproducible transcript expression between all variants[37]. To measure eclosion, homozygous *da-GAL4* virgin females were crossed to heterozygous *UAS-PTEN variants/TM3, Ser, Sb,GMR-Dfd-EYFP* (TDGY) males at a ratio of 2:1. These adults are predicted to produce progeny of genotype *da-GAL4/UAS-PTEN* and *da-GAL4/TDGY* at a 50:50 Mendelian ratio.

In this way, the *da-GAL4/TDGY* flies serve as an internal genetic control (that is the same in every vial examined) to which the developmental rate of *da-GAL4/UAS-PTEN* variants can be compared. For every PTEN variant tested, 3 vials of mating flies were set up and combined time to eclosion rates was measured between vials for both PTEN and TDGY progeny. For every batch of PTEN variants that were inserted in parallel at the same time, we also set up a three vials each of *da-GAL4* crossed to *WT/TDGY*, *attP2/TDGY* and *C124S/TDGY*. Mating flies were maintained on standard cornmeal food at 25 °C and 70% humidity at a constant 12 h light/dark cycle. Prior to the assay, mating flies were housed for a 36–48-h acclimatization period. For the assay, mating flies were placed on fresh yeast-augmented food for 1 h to clear out older fertilized eggs, then placed in a fresh vial in which females laid freshly fertilized eggs within a 6-h window. Adult flies were removed from this vial and progeny were monitored until eclosion. The appearance of the first adult fly in each vial was considered as time zero, and adult fly eclosion was assessed thereafter every 12 h for the next 3 days, comparing the time to eclosion for EYFP-positive and EYFP-negative flies per vial. To control for any impact of the genetic background of the *da-Gal4 driver*, we confirmed that we obtained similar time to eclosion data for over-expression of WT-PTEN, or the LoF variant D326N, regardless of whether driven from either *actin-GAL4, tubulin-GAL4* driver, or *da-GAL4*.

**Neuronal growth in rat primary neuronal cultures.** All experimental procedures and housing conditions were approved by the University of British Columbia Animal Care Committee and were in accordance with the Canadian Council on Animal Care (CCAC) guidelines. Dissociated hippocampi and dorsal root ganglia (DRG) were isolated from embryonic day 18.5 (E18) of Sprague-Dawley rats (Charles River, Sherbrooke, Canada) of either sex prepared as previous described[67] and plated at a density of 130 cells/mm$^2$ for hippocampus, and 50 K per coverslip pre-coated with poly-D-lysine for DRG. Neurons were co-transfected with enhanced-GFP (eGFP) and *PTEN* variants with Lipofectamine 2000 (Invitrogen/Life Technologies, CA, USA) at 9-11 days in vitro (DIV) for hippocampus, and 1 DIV for DRG. Each *PTEN* variant was tagged with three repeats of an HA epitope tag (amino acid sequence: YPYDVPDYA) with a 24 amino acid linker at the N-terminus (MYPYDVPDYAYPYDVPDYAYPYDVPDYASTARRILQVITSLYK KAEGPSRPT), and expression was driven by a pCAG promoter. Hippocampal cultures were fixed at 13-14 DIV for measures of synaptogenesis and morphogenesis, and DRG were fixed at 24 h after transfection for measures of axonal growth. For immunocytochemistry, cells were incubated with the primary antibodies mouse anti-PSD-95 (1:500, Abcam, cat.#ab2723), or mouse anti-Gephyrin (1:300, Synaptic Systems, cat.#147 011) overnight at 4 °C in 1% goat serum/PBS, and incubated with Alexa488 or Alexa568 conjugated goat anti-mouse secondary antibodies (1:1000, Molecular Probes, cat.# A-11001, cat.#A-11019) for 1 h at room temperature. Neurons were mounted on microscope slides with Prolong Gold (Molecular Probes, Thermo Fisher Scientific, MA, USA) and imaged using an Olympus Fluoview 1000 confocal microscope. For DRG, antibodies used were anti-Beta-Tubulin III (1:500, STEMCELL Technologies, clone TUJ1, cat.#60052) and Texas Red dye-conjugated AffiniPure Goat Anti-Mouse IgG (1:250, Jackson ImmunoResearch Laboratories Inc., cat.#111-075-144), mounted with VECTA-SHIELD® Antifade Mounting Media. Dendritic morphology and soma size were imaged using a 20×/0.75 Oil Plan-Apochromat objective. To quantify synaptic density, fixed neurons immunostained with antibodies against Gephyrin (inhibitory synapses) and PSD-95 (excitatory synapses). Synapse were imaged using a ×60/1.42 Oil Plan-Apochromat objective. Synaptic puncta were defined as 0.05–3 μm, and synaptic density was calculated by obtaining the total number of puncta within the dendrite divided by the total dendrite length. Puncta numbers were obtained using a custom macro that made use of the Analyze Particles function of ImageJ and the Colocalization plugin (https://imagej.nih.gov/ij/plugins/colocalization.html), and total length was measured using ImageJ[68]. Images of neurons were obtained using a Zeiss Axioplan II fluorescence microscope equipped with a Plan-NEOFluar ×20/0.50 objective lens. For each culture, 10 GFP-positive neurons were randomly sampled. To evaluate neurite length and branching, neurites (≥2 μm) were traced using ImageJ 1.52a software equipped with the NeuronJ plugin.

**C. elegans chemotaxis assay.** Worms were cultured on Nematode Growth Medium (NGM) seeded with *Escherichia coli* (OP50). N2 Bristol, and CB1375 *daf-18(e1375)* strains were obtained from the *Caenorhabditis* Genetics Center (University of Minnesota, USA). All non-tagged *PTEN* variants were expressed as pan-neuronal extrachromosomal arrays injected into the *daf-18(e1375)* reduction-of-function background at 50 ng/μl final DNA concentration. *daf-18(e1275)* harbors a 30–base pair insertion in the fourth exon and is predicted to insert six amino acids before introducing an early stop codon that truncates the C-terminal half of the protein while leaving the phosphatase domain intact. The strains created for this work are listed in Supplementary Table 1. The *PTEN* entry clones were recombined with a pDEST-*aex-3p* destination vector for pan-neuronal expression (obtained from Dr. Hidehito Kuroyanagi, Tokyo Medical and Dental University) to generate the *aex-3p::PTEN::unc-54 UTR* rescue construct using gateway cloning (Invitrogen), according to manufacturers instructions. Multiple transgenic strains were generated per genotype using standard microinjection and fluorescent screening for a *pmyo-2::mCherry::unc-54 3'UTR* (pCFJ90) co-transformation marker

(3 ng/ul). *PTEN* variant sequences were confirmed by amplifying the entire CDS PCR followed by Sanger sequencing. The forward and reverse primers used to amplify the *PTEN* CDS were ATGACAGCCATCATCAAAGA and TCAGACTTTTGTAATTTGTG respectively. The chemotaxis behavioral assay was conducted on a 6-cm assay plate (2% agar) where a salt gradient was formed overnight by inserting a 2% agar plug containing 50 mM of NaCl (~5 mm in diameter) 1 cm from the edge of the plate[69]. A control 2% agar plug without NaCl was inserted 1 cm from the opposite edge of the plate. Strains were grown on NGM plates seeded with *E. coli* (OP50) for 3 or 4 days. Worms on the plates were collected and washed three times using M9 buffer before being pipetted onto an unseeded NGM plate to remove excess buffer and select animals carrying transformation markers. Adult worms were transferred and placed at the center of the assay plates and tracked for 40 min on the Multi-Worm Tracker[47]. After the tracking period, the chemotaxis index was calculated as $(A - B)/(A + B)$, where A was the number of animals that were located in a 1.5-cm-wide region on the side of the assay plate containing the 2% agar plug with 50 mM NaCl and B was the number of animals that were located in a 1.5-cm-wide region on the side of the assay plate containing the 2% agar plug without NaCl. Animals not located in either region (ie. the middle section of the assay plate) were not counted towards the chemotaxis index. One hundred to two hundred animals were used per plate, and two or three plate replicates were used for each line in each experiment. Multi-Worm Tracker software (version 1.2.0.2) was used for image acquisition[47]. Behavioral quantification with Choreography software (version 1.3.0_r103552) used --shadowless, --minimum-move-body 2, and --minimum-time 20 filters to restrict the analysis to animals that moved at least 2 body lengths and were tracked for at least 20 s. Custom R scripts organized and summarized Choreography output files. Each experiment was independently replicated at least twice. No blinding was necessary because the Multi-Worm Tracker scores behavior objectively.

**Variant protein abundance measures in yeast.** Yeast were grown overnight under inducing conditions. $A_{600}$ was measured for all samples then each were diluted to an optical density (OD) of 0.1 and allowed to grow ~6–8 h or until they reached between 0.8 and 1.0 OD. An equivalent of 1.0 OD was collected for each sample. Samples were pelleted and supernatant discarded before being frozen at −20 °C. After freezing, glass beads and sample buffer containing sodium dodecyl sulfate (SDS) and 2.0% β-mercaptoethanol were added to the pellet then bead bashed for 2 min. Samples were then boiled at 90 °C for 5 min and centrifuged before being loaded into a 4–12% Bis-Tris protein gel (Thermo Fisher Scientific, cat #NP0322) and run with MOPS running buffer (Thermo Fisher Scientific, cat. #NP0001). Gels were transferred in buffer containing 20% methanol and 0.03% SDS for 2 h at 4 C at 28 V onto 0.2 μM nitrocellulose membranes (Bio-Rad, cat. #1620097). Membranes were blocked in tris-buffered saline and Tween 20 (TBST) (Acros Organics, cat #AC233360010) containing 5% milk (Bio Basic, cat. #NB0669). Membranes were then stained with PTEN antibody (1:1000; R&D Systems, MAB847) followed by goat-anti-mouse HRP (1:5000, Thermo Fisher Scientific, cat. #62-6520) and imaged with ChemiDoc MP (Bio-Rad). Membranes were then stripped with a pH 2.2 stripping buffer containing 1.5% glycine, 0.1% SDS and 1% Tween 20 and re-stained for the loading using the anti-glucose-6-phosphate dehydrogenase antibody (Millipore, cat. #A9521). Blots were again imaged on ChemiDoc MP (Bio-Rad). Band density was determined using ImageJ. All PTEN bands were normalized first to their loading control then the wild-type PTEN done in each experiment. All variants were tested in triplicate.

**HEK293 cell culture, stability, and pAKT/AKT assays.** HEK293 cells purchased from the American Type Culture Collection (ATCC, cat. #CRL-1573) and were routinely passaged in Dulbecco's Modified Eagle's Medium (DMEM) (MilliporeSigma, cat. #D6046) supplemented with 10% FBS and 100 U/mL Penicillin-Streptomycin. For all experiments herein, HEK293 cells were used for a maximum of 15 passages. Cells were seeded at $1 \times 10^5$ per well in 24-well dishes 16–20 h before transfected with 500 ng of expression plasmid using X-tremeGENE 9 at a ratio of 2 μL to 1 μg DNA. All variants were expressed as an N-terminal fusion with sfGFP with a 13 amino acid linker (TSLYKKAGSEFAL), and expression was driven by a pCAG promoter. In all, 24 h after transfection cells were washed with DMEM and starved for 19 h in serum-free media before being stimulated for 10 min with complete medium plus 10 nM Insulin (Gibco, cat. #12585014), then washed once in 1xPBS before treated with Trypsin-EDTA (Gibco, cat. # 25200072) for 5 min to create a single-cell suspension and then fixed for 10 min in 3.2% PFA. Cells were permeabilized in Flow Cytometry Permeabilization/Wash Buffer I (FC005, R&D Systems) and stained with Rabbit anti-pAKT (1:100; Cell Signaling, Ser473, cat. #9271) and Mouse anti-pan-AKT (1:100; Cell Signaling, cat. #2920,) for 1 h on ice. Cells were washed and then stained with Goat anti-Rabbit IgG-Alexa Fluor 647 (1:100; Thermo Fisher, cat. #A21244) and Goat anti-Mouse IgG-Alexa Fluor 405 (1:100; Thermo Fisher, cat. #A31553) for 1 h on ice. Cells were washed and re-suspended in Flow Cytometry Staining Buffer (FC001, R&D Systems) before loading into an Attune Nxt Flow Cytometer (Invitrogen). Data was recorded using VL-1 (Alexa Fluor 405), BL-1 (sfGFP), YL-1 (mtagRFP-T), and RL-1 (Alexa Fluor 647) channels, which were single-stain compensated. Using FlowJo, Cells were selected using FSC-H/SSC-H and single cells were selected using SSC-H/SSC-A. For pAKT/AKT measures, cells with sfGFP fluorescence above untransfected cells to 100-fold above untransfected (300 a.u. to 20,000 a.u.) were selected. For stability

measures, cells with higher mtagRFP-T fluorescence than untransfected and sfGFP fluorescence <20,000 a.u. were selected. The median of (647 - Background)/(405 - Background) as the pAKT/AKT level was calculated both for the positive and negative (RFP- and sfGFP-, untransfected) population in each sample and the difference was measured as the effect measure, normalizing each sample by its in-well untransfected control. Samples were then normalized to WT = 1 for each day of experiment. For stability measures, the median ratio of (sfGFP - Background)/(mtagRFP-T – Background) was calculated as the relative stability of PTEN compared to its 1:1 transfection control. Samples were then normalized to WT = 1 for each day of experiment.

**Data modeling and variant effects analysis.** The quantitative phenotypes for the yeast, fly, worm, and rat assays were analyzed using a hierarchical (mixed effect) model approach to account for experimental batch effects and variability between replicates. For each assay, we treat the genotype as a fixed effect, which can be either the positive control, the negative control, or a variant, and treat blocking factors such as experimental batch or day as random effects (stratified by variant and accompanying controls). All R scripts are provided at https://github.com/PavlidisLab/Post-PTEN. For each assay, we fit models to the data for all tested variants and accompanying controls jointly. Given a genotype $i$ and a sample $j$, we fit:

$$y_{i,j} = X_{i,j}\beta_{i,j} + Z_{i,j}u_{i,j} + \varepsilon$$

Where $y$ is the observed quantitative phenotype (length $N$ where $N$ is the number of data points for the assay in total), $X$ and $Z$ are the model matrices for fixed and random effects respectively, is a vector of parameters for the fixed effects (the genotypes of each variant), $u$ is vector of parameters for random effects, and $\varepsilon$ is residual error. Models were fit using the lmer function of the lmerTest package[70] in the R programming environment (R Core Team 2019). To assess the significance of effect between the positive (wild-type PTEN) and negative (no PTEN, empty vector or GFP background) controls, we extracted the model fit $p$-values of each genotype under a comparison where each control is used as the contrast group, using the method of Satterthwaite as implemented in lmerTest. Both comparison yields a vector of $p$-values for all other genotypes except the contrast variant. For visualization purposes, we plot data adjusted for the estimated random effects. Specifically, we compute adjusted values through a multiplication between the fixed-effects model matrix X with the β fixed-effect parameter estimate and a sum of the model residuals vector. In other words:

$$\widehat{y}_{i,j} = X_{i,j}\widehat{\beta}_{i,j} + \widehat{\varepsilon}$$

We then take the $y$ values for each genotype and rescale the data such that the mean positive control equals 1.0, and the mean negative control equals 0.0. Finally, we rescale the standard errors for each genotype from the adjusted data proportionally to the ratio of the adjusted data and the normalized data distributions, thus keeping the values consistent with the 0.0–1.0 re-scaling. All results are presented as means ± SEM. For HEK293 experiments, data were analyzed with Pearson correlations tests and Student's $t$ tests with WT unless otherwise indicated. All statistical analysis was done using the statistical programming software Graphpad $p \le 0.05$ was considered as significant.

**Reporting summary.** Further information on research design is available in the Nature Research Reporting Summary linked to this article.

## Data availability

The datasets generated during and/or analyzed during the current study are available at https://doi.org/10.5683/SP2/DQOKPB. The source data underlying Figs. 1–8, and Supplementary Figs. 1–6 are provided as a Source Data file.

This work makes use of the following publicly available databases:
VariCarta1 (https://varicarta.msl.ubc.ca/index)
SFARI Gene (https://gene.sfari.org/)
ExAC (http://exac.broadinstitute.org/)
gnomAD v2.1.1 (https://gnomad.broadinstitute.org/)
CADD phred version 1.0 31 (https://cadd.gs.washington.edu/)
SNAP2 (https://www.rostlab.org/services/snap/)
TransVar (https://bioinformatics.mdanderson.org/transvar/)
RefSeq (https://www.ncbi.nlm.nih.gov/refseq/)
Annovar (https://doc-openbio.readthedocs.io/projects/annovar/en/latest/)
wAnnovar (http://wannovar.wglab.org/)

## Code availability

All code used for data modeling is provided at: https://github.com/PavlidisLab/Post-PTEN

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

## Acknowledgements

This work was supported by grant from the Simons Foundation (SFARI award #573845, Grantees: KH, PP, DWA, CJL, SXB, TPO, CHR), and a CIHR Foundation Award (KH).

## Author contributions

K.L.P., M.B., P.G., and F.M. contributed equally as first-authors. R.D., T.A.M., and W.M.M. contributed equally as second authorship position. Variant identification and classification were carried out by K.L.P., M.B., D.B.C., S.R., and P.P. Variant library construction was conducted by F.M., A.N., K.P., W.M.M., and K.H. Yeast assays were conducted by K.L.P., B.P.Y., and C.J.L. *Drosophila* assays were conducted by P.G. and D.W.A. Rat neuronal assays were conducted by R.D., M.E., C.H., T.O.C., and S.X.B. *C. elegans* assays were conducted by T.A.M. and C.H.R. HEK293 assays were conducted by F.M., W.M.M., and K.H. The HEK293 KO PTEN cell line was constructed by A.C., F.M., W.M.M., and K.H. Statistical analyses were performed by M.B., F.M. and P.P. The manuscript was written by K.L.P. and K.H. and edited by F.M. and W.M.M. All authors approved manuscript prior to submission.

## Competing interests

The authors declare no competing interests.
