## [Peer Review File · Nature Communications]

Reviewers' Comments:

Reviewer #1:

Remarks to the Author:

The manuscript by Post et al. is an ambitious, comprehensive and powerful example of deep phenotyping based on establishing physiologically relevant assays in multiple organisms and cell systems. Mutations of PTEN, predominantly found in ASD, but also in PHTS and cancer, were systematically assessed for their ability to impact phenotypes in yeast, fly, worm, rat cortical neurons and HEK293 cells. The results of the study represent an important resource for future PTEN research, management of diseases associated with PTEN alterations and add to the map of PTEN regions implicated in maintenance of protein stability and enzymatic activity. While much of the work has been well-executed, described and interpreted, a few issues should be addressed to justify publication.

1. The choice of *S. cerevisiae* as a model system when assessing PTEN function is dubious, as yeast do not have a functional PTEN homolog nor its preferred substrate, PI(3,4,5)P₃, as the authors acknowledge in the text. Lack of this "endogenous framework" likely skews the presented results towards non-specific genetic interactions and non-physiological associations. While I appreciate the throughput of the yeast assay, it should be shown how many of the "relevant" interactions identified by the correlative analyses in the latter parts of the manuscript hold without the inclusion of yeast data. The interactions identified independent of the yeast data should be held in higher biological regard and separately/prominently reported in the text.
2. Further explanation of the use of PTEN fusion proteins needs to be provided, as the fusions may impact PTEN activity/stability. From the figure legends, it appears that majority of HEK293 experiments were performed with GFP-PTEN fusions, yet no detailed relation of this reagent to PTEN is provided. Does it perform equivalently to PTEN without the fusion(s)? Have any of the results from the "stability" mutants been validated in direct stability assays (pulse-chase or similar) and without the GFP fusion? This data needs to be included, at least for a subset of phenotypically most prominent mutants. Also, any use of PTEN fusions and their possible limitations in the other parts of the manuscript should be fully documented.
3. The interpretation/discussion of the potential "dominant-negative" PTEN mutants should be better developed. While there exists literature on PTEN "dominant-negative" mutants, as judged by the downstream effects on the Akt pathway, the mechanistic aspects of the function of those PTEN mutants are far from clear. In fact, the very notion of a dominant-negative phosphatase, any phosphatase, is conceptually flawed. Unlike dominant-negative protein kinases, which interfere with the physiological phosphorylation of their substrates by virtue of blocking access to the substrate by the present endogenous wt kinase, the presumed "dominant-negative" phosphatases can simplistically be understood as de novo phosphate-binding domains. Depending of the further accessibility of the target phosphate for further signaling, "dominant-negative" phosphatases can have both positive and negative effects on signal propagation. It is thus advisable that the presented report divest from the presented interpretation of dominant-negative outputs, as they don't add any further value to the reported data.
4. The "Assessment of predictive algorithms" section is not fully realized and could be the beginning of a separate body of work. Without functional cross-validation between platforms, it is difficult to compare the outputs of the presented algorithms. I suggest that this section be removed, as it will not reduce the impact of the presented work.

Vuk Stambolic

Reviewer #2:

Remarks to the Author:

"Multi-model functionalization of disease-associated PTEN missense mutations identifies multiple molecular mechanisms underlying protein dysfunction", is a clearly written and impactful manuscript. The manuscript represents a tremendous effort to examine PTEN structure/function in

an unbiased fashion. Specifically, the manuscript describes the impact of an exhaustive list of autism and cancer-associated point mutations on protein stability and function. The authors first demonstrate a functional assay for Pten using yeast, then *Drosophila*, rat primary neurons, and *C. elegans*. This impressive multi-model system approach allows the authors to draw conclusions about Pten that are not likely an artifact of strengths and weaknesses of any one model system. Ultimately, this work demonstrates validity of using such an approach on disease associate genes in which structure/function is unknown.

Pten has been heavily studied in the context of cancer and neurodevelopment and function. The multi model approach in this system is validated by readily identifying dysfunction in mutants found in known catalytic domains of Pten. Further, the different models allowed extensive analysis of protein stability. My one minor concern with the approach is the lack of discussion about detecting loss of function for mutations found in the C2 domain and C-Tail. These regions are largely thought to regulate subcellular localization and access to catalytic substrate. Therefore, the ability of the different model systems to detect loss of function based on protein trafficking and substrate contact should be considered in the discussion.

Reviewers' comments:

Reviewer #1 (Remarks to the Author):

The manuscript by Post et al. is an ambitious, comprehensive and powerful example of deep phenotyping based on establishing physiologically relevant assays in multiple organisms and cell systems. Mutations of PTEN, predominantly found in ASD, but also in PHTS and cancer, were systematically assessed for their ability to impact phenotypes in yeast, fly, worm, rat cortical neurons and HEK293 cells. The results of the study represent an important resource for future PTEN research, management of diseases associated with PTEN alterations and add to the map of PTEN regions implicated in maintenance of protein stability and enzymatic activity. While much of the work has been well-executed, described and interpreted, a few issues should be addressed to justify publication.

We thank the Reviewer for this strong endorsement for the manuscript.

1. The choice of *S. cerevisiae* as a model system when assessing PTEN function is dubious, as yeast do not have a functional PTEN homolog nor its preferred substrate, PI(3,4,5)P₃, as the authors acknowledge in the text. Lack of this “endogenous framework” likely skews the presented results towards non-specific genetic interactions and non-physiological associations. While I appreciate the throughput of the yeast assay, it should be shown how many of the “relevant” interactions identified by the correlative analyses in the latter parts of the manuscript hold without the inclusion of yeast data. The interactions identified independent of the yeast data should be held in higher biological regard and separately/prominently reported in the text.

We thank the Reviewer for this important input and appreciate their concerns. We were initially similarly cautious about using yeast as a model system for assessing human PTEN variant function, but 4 lines of evidence now make us more confident of its utility.

1) We were surprised and encouraged by the results from our unbiased SDL screen testing for genetic interactions, which identified 8 genes that interact with wt human PTEN, all of which play direct or indirect roles in phospholipid metabolism. For example, Vac7 and Vac14 regulate synthesis of PI(3,5)P₂, and Fig4 dephosphorylates PI(3,5)P₂ to form PI3P. Vam3 and Vam7 are SNARE proteins and YPT7 is a Rab GTPase, all three are components of the vacuolar fusion complex, and Vam7 binds PI3P on the vacuolar membrane to initiate fusion. Moreover, the PTEN variant interactions for each of these 8 genes are well correlated across genes (Supp. Fig. 1)

2) While yeast does not contain the primary PTEN substrate, PIP₃, PTEN also binds PI3P as a secondary substrate, which is present in yeast and human cells. Thus, we predict that our yeast growth assay is sensitive to a physiologic process, with results potentially skewed toward features that are specific to secondary substrate actions. We discuss this in the Discussion and we elaborate on potential specific sensitivities of different yeast sentinel strains to different variants, which we believe adds strength to their utility for dissecting molecular mechanisms of protein dysfunction.

3) Reassuringly, the 'Biochemical Variants' C124S, G129E, Y138L, and C124S-4A are mutations that have been well studied in other model systems and performed as expected in yeast.

4) Critically, our PTEN variant measures of stability/abundance and function in yeast are highly correlated with measures from our model systems that endogenously express PTEN. We present these direct comparisons of both stability/abundance measures from yeast and HEK293 (Fig. 4e), and functional measures between yeast and HEK293 (Fig. 5e) and yeast and fly (Fig. 2d), for each variant for readers to best critique measures across models. We further show over-all correlations between models (Fig. 6e,f) demonstrating strong correlation between yeast and assays with clearer relevance to human physiology. We interpret these results to indicate that our yeast assay is sensitive to mechanisms regulating stability and function of PTEN variants largely similar to when they are expressed in cellular environments expressing endogenous PTEN.

The only component of our study which aggregates data across models is our assessment of pathogenicity (Fig. 8b) which is based on whether variant dysfunction is observed in one or more models. To aid readers' critique of these measures, we directly provide heatmaps for variant impact in each model for each variant (Fig. 8a,b). Given these evidence described above, we feel that the benefit of adding another sensitive measure of PTEN variant stability and function warrants inclusion of yeast data in our pathogenicity metric. It should be noted, that we do not use all 8 sentinels in our metric – we only use data from the Vac14 sentinel (since it has the largest dynamic range for sensitivity), to not unduly bias the metric towards the yeast model.

2. Further explanation of the use of PTEN fusion proteins needs to be provided, as the fusions may impact PTEN activity/stability. From the figure legends, it appears that majority of HEK293 experiments were performed with GFP-PTEN fusions, yet no detailed relation of this reagent to PTEN is provided. Does it perform equivalently to PTEN without the fusion(s)? Have any of the results from the "stability" mutants been validated in direct stability assays (pulse-chase or similar) and without the GFP fusion? This data needs to be included, at least for a subset of phenotypically most prominent mutants. Also, any use of PTEN fusions and their possible limitations in the other parts of the manuscript should be fully documented.

We thank the Reviewer for pointing out our oversight in clarifying tagging of PTEN variants in the different models systems we employed. Non-tagged PTEN variants were used for yeast, Drosophila and C. elegans experiments, and sfGFP-fusions and 3XHA-tags were used for HEK293 cells and rat neuronal cultures, respectively. We have added new text in the Results, Figure Legends and Methods clarifying when non-tagged or tagged proteins were employed.

We share with the Reviewer's concerns regarding measuring PTEN variant stability using a GFP-fusion strategy. Fortunately, for PTEN this problem has been addressed in a recent publication from the Fowler lab (Matreyek, et al., Nature Genetics, 2018: PMID: 29785012) in which they find similar stabilities for PTEN variants fused to full-length GFP and a 15-amino acid split GFP tag (Matreyek, et al., Supp. Fig. 1d), as well as compared to untagged PTEN variants (Matreyek, et al., Supp. Fig. 3c, Supp. Table 9). We have added new text highlighting these concerns and the Fowler lab findings.

In our study, we find support for a lack of influence of the GFP-tag on PTEN variant stability by the strong correlation between stability of GFP-fusion PTEN variant in HEK293 cells and abundance measures of untagged PTEN variants in yeast (Fig. 4e). Moreover, we see strong correlation between variant functional measures in HEK293 (GFP-fusions) with functional measures using non-tagged PTEN variants in both yeast and fly (Fig. 5e,f). We have added new text discussing our results indicating lack of detectable impact of GFP-fusion or HA-tag on PTEN stability and function.

3. The interpretation/discussion of the potential “dominant-negative” PTEN mutants should be better developed. While there exists literature on PTEN “dominant-negative” mutants, as judged by the downstream effects on the Akt pathway, the mechanistic aspects of the function of those PTEN mutants are far from clear. In fact, the very notion of a dominant-negative phosphatase, any phosphatase, is conceptually flawed. Unlike dominant-negative protein kinases, which interfere with the physiological phosphorylation of their substrates by virtue of blocking access to the substrate by the present endogenous wt kinase, the presumed “dominant-negative” phosphatases can simplistically be understood as de novo phosphate-binding domains. Depending of the further accessibility of the target phosphate for further signaling, “dominant-negative” phosphatases can have both positive and negative effects on signal propagation. It is thus advisable that the presented report divest from the presented interpretation of dominant-negative outputs, as they don’t add any further value to the reported data.

We agree that the mechanism(s) by which PTEN variants elicit dominant negative (DN) effects are poorly understood. We had removed discussion of this topic in the original manuscript submission due to space limitations, but agree on its importance and have added new text to address this issue. We describe 3 potential mechanisms: substrate sequestration, dimerization-mediated inhibition of wt, and substrate switching.

We agree with the Reviewer that substrate sequestration is possible to occur, but unlikely given our results (and those of others) since, as stated, it could have ‘both positive and negative effects on signal propagation’. However, we realize that we cannot fully discount this possibility.

Rather, we believe our data best fits a dimerization-mediated inhibition model as described by Herskowitz (Nature, 1987; PMID: 2442619), in which heterodimerization of the wt protein with a dysfunctional variant reduces function of the bound wt. Papa, et al. (Cell, 2014; PMID: 24766807) indicates that wt PTEN homodimerization-mediated enhancement of lipid phosphatase activity is reduced in heterodimers of wt with the PTEN variants C124S and G129E. Our results with C124S and G129E replicate their findings. In addition, we identified 25 previously undescribed PTEN variants with DN effects on pAKT. Critically, we created a PTEN KO HEK293 cell line using CRISPR/Cas9, 6B1, to validate DN by demonstrating loss of this effect in the majority of these variants.

We also discuss a third potential alternative to DN, substrate switching, a recently reported mechanism in which a mutation within the P-loop changes PTEN from a 3- to 5-phosphatase yielding catalytic products resulting in enhanced, rather than reduced AKT function (Costa, et al. PNAS, 2015; PMID: 26504226). Given that such a mechanism would not explain the loss of DN activity when variants are

expressed in PTEN KO cells, we believe substrate switching is unlikely a mechanism for the majority of the DN behavior we observe. However, substrate switching might explain why a few variants, including A126D, retain DN effects in the PTEN KO cell line.

We agree on the importance of this line of investigation, but recognize that besides our efforts to create and apply our PTEN KO cell line, further exploration of mechanisms underlying DN effects are well beyond the scope of the current study. We believe that our identification of 25 novel DN variants is a significant contribution to these future studies.

4. The “Assessment of predictive algorithms” section is not fully realized and could be the beginning of a separate body of work. Without functional cross-validation between platforms, it is difficult to compare the outputs of the presented algorithms. I suggest that this section be removed, as it will not reduce the impact of the presented work.

We agree with the Reviewer and have removed this section from the manuscript. We have also removed Supplemental Fig. 7, and CADD/SNAP2 measures from Fig. 6e,f, Supp. Fig. 3a,b, and Supp. Fig. 5a,b.

Reviewer #2 (Remarks to the Author):

“Multi-model functionalization of disease-associated PTEN missense mutations identifies multiple molecular mechanisms underlying protein dysfunction”, is a clearly written and impactful manuscript. The manuscript represents a tremendous effort to examine PTEN structure/function in an unbiased fashion. Specifically, the manuscript describes the impact of an exhaustive list of autism and cancer-associated point mutations on protein stability and function. The authors first demonstrate a functional assay for Pten using yeast, then Drosophila, rat primary neurons, and C. elegans. This impressive multi-model system approach allows the authors to draw conclusions about Pten that are not likely an artifact of strengths and weaknesses of any one model system. Ultimately, this work demonstrates validity of using such an approach on disease associate genes in which structure/function is unknown.

We are encouraged by the enthusiasm of Reviewer #2 for the importance of our work.

Pten has been heavily studied in the context of cancer and neurodevelopment and function. The multi model approach in this system is validated by readily identifying dysfunction in mutants found in known catalytic domains of Pten. Further, the different models allowed extensive analysis of protein stability. My one minor concern with the approach is the lack of discussion about detecting loss of function for mutations found in the C2 domain and C-Tail. These regions are largely thought to regulate subcellular localization and access to catalytic substrate. Therefore, the ability of the different model systems to detect loss of function based on protein trafficking and substrate contact should be considered in the discussion.

We thank the Reviewer for this input, and we agree that even with our diverse multi-model approach we do not fully assess all functions of PTEN. While the N-terminal substrate binding and catalytic domains exhibited conserved stability-independent sensitivity across our models, variants in the C-terminal region

encompassing the C2 domain, serine/threonine-rich C-tail and PDZ domain largely exhibited stability-dependent function, indicating these regions are important to confer protein stability, in addition to their well established role in regulating PTEN activity. We recognize that our failure to detect stability-insensitive sites or domains in these regions is likely due to the sensitivity of the specific assays employed, particularly given that phosphorylation of C2 and the C-tail inhibits its membrane association (reviewed by Ross and Gericke, 2009, PNAS; PMID: 19174524), and variants within the C2 domain have been shown to exhibit LOF due to their reduced nuclear localization (Fricano-Krugler et al., 2018; PMID: 29373119). We have added new text to the Discussion to address this concern.

Reviewers' Comments:

Reviewer #1:

Remarks to the Author:

The authors have addressed the comments and the manuscript is acceptable for publication.

Reviewer #3:

Remarks to the Author:

Post et al. perform a series of assays on PTEN variants associated with human disease in 5 model systems. These variants include null, hypomorphic, neomorphic, and perhaps antimorphic alleles of PTEN and find a positive correlation in their impacts across the various models used. They also find assay-specific effects, likely reflective of specific functions in the particular systems and tissue types being studied. These results will be of impact and the authors provide a good foundation to justify the interest in assessing PTEN variants in yeast (despite lacking a PTEN homolog and PIP3), flies, worms, rodents, and HEK cells. The statistical analyses are appropriate. Overall, these series of multi-model insights provide an important and systematic demonstration of PTEN variant functions that illuminate the complexities of single point mutations, defects in protein function, and shared as well as distinct roles in complex systems and tissues.

Some points to consider:

1. The use of untagged PTEN variants expressed in *Drosophila* and *C. elegans* is a good approach to minimize potential confounds of N- or C-terminal tags. Even better, targeting the same attP2 insertion site allows for controlled comparison between variants by maintaining equivalent expression levels. One question is the choice of da-GAL4 as the only driver assayed for the overexpression of PTEN variants in modulating developmental rate. Although daughterless is thought to be ubiquitously expressed, it is possible that the driver or genetic background may have peculiarities that bias the impact PTEN variants have on developmental rates. While it would not be reasonable to ask that all the assays be performed with a separate ubiquitous driver (tubulin, actin, etc), did the authors compare the developmental rates with a few of the PTEN variants crossed to an independent ubiquitous driver before settling on da-GAL4? This concern is tempered by the fact that the authors normalized everything to heterozygous UAS-PTEN variants/balancer, but at minimum the exclusive use of daughterless-GAL4 should be justified.

2. For the *C. elegans* chemotaxis assay, the unique advantage of being able to survive null mutations in PTEN is a notable strength. Although in this system expression levels of PTEN variants cannot be rigorously controlled because of the use of varying extrachromosomal arrays, the expression of all variants in the same daf-18 mutant background and restoration of normal chemotaxis by WT expression of human PTEN provides a well controlled background to assess changes in the behavior. The importance of having a chemotaxis assay, which may have relevance to ASD variants, is well justified in this study.

RESPONSE TO REVIEWERS

Reviewer #3:

Post et al. perform a series of assays on PTEN variants associated with human disease in 5 model systems. These variants include null, hypomorphic, neomorphic, and perhaps antimorphic alleles of PTEN and find a positive correlation in their impacts across the various models used. They also find assay-specific effects, likely reflective of specific functions in the particular systems and tissue types being studied. These results will be of impact and the authors provide a good foundation to justify the interest in assessing PTEN variants in yeast (despite lacking a PTEN homolog and PIP3), flies, worms, rodents, and HEK cells. The statistical analyses are appropriate. Overall, these series of multi-model insights provide an important and systematic demonstration of PTEN variant functions that illuminate the complexities of single point mutations, defects in protein function, and shared as well as distinct roles in complex systems and tissues.

Response: We appreciate the Reviewer's support for the importance of our work.

Some points to consider:

1. The use of untagged PTEN variants expressed in *Drosophila* and *C. elegans* is a good approach to minimize potential confounds of N- or C-terminal tags. Even better, targeting the same attP2 insertion site allows for controlled comparison between variants by maintaining equivalent expression levels. One question is the choice of da-GAL4 as the only driver assayed for the overexpression of PTEN variants in modulating developmental rate. Although daughterless is thought to be ubiquitously expressed, it is possible that the driver or genetic background may have peculiarities that bias the impact PTEN variants have on developmental rates. While it would not be reasonable to ask that all the assays be performed with a separate ubiquitous driver (tubulin, actin, etc), did the authors compare the developmental rates with a few of the PTEN variants crossed to an independent ubiquitous driver before settling on da-GAL4? This concern is tempered by the fact that the authors normalized everything to heterozygous UAS-PTEN variants/balancer, but at minimum the exclusive use of daughterless-GAL4 should be justified.

Response: Prior to settling on the assay used in this manuscript, we screened through a number of potential assays with multiple GAL4 drivers in multiple tissues. Prior to determining that we would use daughterless-GAL4 (da-GAL4) for our large study, we also tested widely-used actin-GAL4 and tubulin-GAL4 drivers in this same time-to-eclosion assay. We found that both GAL4's gave the same results; WT-PTEN delayed the time to eclosion, and loss-of-function variant D326N eclosed at the same time as internal controls. Therefore, the time-to-eclosion assay gave the same results regardless of the GAL4 driver used. We selected da-GAL4 for our assay because of the three GAL4's we worked with, only da-GAL4 is homozygous viable. Thus, once UAS-PTEN^{var} strains were first balanced after transgenesis, as heterozygotes over a TM3,P{Dfd-GMR-nvYFP},Sb¹ balancer chromosome (a standard step in stabilizing the integrated chromosome lines after transgenesis), we could quickly cross them to a homozygous GAL4 line for our assays (that therefore examined eclosion rate of a 50% da-GAL4/UAS-PTEN^{var} vs 50% da-GAL4/TM3,P{Dfd-GMR-nvYFP},Sb¹). Thus, we reduced the alternate progeny genotypes that would have occurred if crossed to e.g. actin-GAL4/CyO. This facilitated a more efficient protocol. In addition, we noted a minor level of lethality of actin-GAL4/+;UAS-PTEN-WT/+

progeny; a confound that we did not wish to introduce into our assay. To aid the reader's interpretation of our data regarding this issue, we have added the following statement to the Methods: "To control for any impact of the genetic background of the da-Gal4 driver, we confirmed that we obtained similar time to eclosion data for over-expression of WT-PTEN, or the LoF variant D326N, regardless of whether driven from either actin-GAL4, tubulin-GAL4 driver, or da-GAL4."

2. For the *C. elegans* chemotaxis assay, the unique advantage of being able to survive null mutations in PTEN is a notable strength. Although in this system expression levels of PTEN variants cannot be rigorously controlled because of the use of varying extrachromosomal arrays, the expression of all variants in the same *daf-18* mutant background and restoration of normal chemotaxis by WT expression of human PTEN provides a well controlled background to assess changes in the behavior. The importance of having a chemotaxis assay, which may have relevance to ASD variants, is well justified in this study.

Response: *The reviewer is right that one of the paramount advantages of our C. elegans assay is that homozygous daf-18 null animals are viable, allowing us to conduct functional assays based on transgenic rescue with untagged human PTEN in vivo. This advantage, coupled with the ability to look at whole-organism sensory behavior as a functional readout was one of the main motivations for the C. elegans experimental design. Thus, we were especially enthusiastic to see that the in vivo rescue-based C. elegans results correlated strongly with the results from the other functional assays.*